# SynDoc: A Hybrid Discriminative-Generative Framework for Synthetic Domain-Adaptive Document Key Information Extraction

## Abstract

Domain-specific Visually Rich Document Understanding (VRDU) presents significant challenges due to the complexity and sensitivity of documents in fields such as medicine, finance, and material science. Although existing Large (Multimodal) Language Models (LLMs/MLLMs) achieve promising results, they still suffer from hallucinations, limited domain adaptation, and heavy reliance on extensive fine-tuning datasets. We introduce SynDoc, a novel framework that combines discriminative and generative models to address these challenges. SynDoc features a robust synthetic data generation workflow that extracts structural information and generates domain-specific queries to produce high-quality annotations. Through adaptive instruction tuning, SynDoc improves the discriminative model's ability to extract domain-specific knowledge. In parallel, a recursive inference mechanism iteratively refines outputs from both models to achieve stable and accurate predictions. This integrated framework demonstrates scalable, efficient, and precise document understanding, bridging the gap between domain-specific adaptation and general world knowledge for key information extraction tasks.

## 1 Introduction

With the increasing demand for domain-specific Visually Rich Document Understanding (VRDU), significant opportunities are emerging across diverse areas such as medicine (Ding et al., 2023b), finance (Ding et al., 2023a), material science (Khalighinejad et al., 2024), and politics (Wang et al., 2023). These areas often rely on documents that contain extensive domain-specific knowledge and sensitive information, which pose unique challenges of key information extraction (KIE). As industries increasingly turn to AI-powered solutions for document analysis, the need for robust and adaptable frameworks to navigate these complexities has never been greater.

Vision-Language Pretrained Models (VLPMs) have demonstrated significant advancements in VRDU, typically in a **discriminative** manner by directly mapping multimodal inputs to key information through classification and sequence labeling (Huang et al., 2022; Gu et al., 2021; Lyu et al., 2024). Yet, these models encounter several challenges. First, they heavily depend on large-scale, domain-specific fine-tuning datasets (Ding et al., 2024a). Second, their performance often degrades in practical, zero-shot scenarios. In contrast, Multimodal Large Language Models (MLLMs) have recently been applied to extract key information from VRDs in a **generative** manner (OpenAI, 2024; Zhu et al., 2025), achieving remarkable progress due to their broad general knowledge; however, MLLMs still lack sufficient target-domain understanding, leading to unreliable or imprecise outputs in VRDU applications. For instance, as shown in Figure 1, an MLLM extracts the *present voting power* "18.86%" instead of the requested *previous voting power*, illustrating its limitations in understanding table structures.

Recent research has explored various strategies to address these challenges in document KIE, with synthetic data generation increasingly emerging as a crucial driver of progress for both discriminative and generative models (Wang et al., 2025; Xie et al., 2025; Tang et al., 2024). In the **discriminative** paradigm, domain-adaptive techniques applied to VLPM backbones have achieved promising results through fine-tuning on curated annotated datasets (Ding et al., 2024a). However, this approach remains constrained by high annotation costs and weak zero-shot performance. In contrast, **generative**

models leverage synthetic data for self-supervised pretraining (Ye et al., 2023; Feng et al., 2024) and instruction tuning (Hu et al., 2024b; Zhang et al., 2025) to enhance multimodal VRD comprehension. Yet, these models face obstacles, including massive computational demands and suboptimal zero-shot performance when deployed in new domains. Moreover, synthetic data generated by MLLMs often suffers from low quality or inconsistency, particularly in question-answer pairs, leaving a gap in research on how to improve the trustworthiness of these synthetically generated instruction-response pairs (Ding et al., 2024b).

In this study, we propose Syn-Doc, a hybrid framework that introduces a new paradigm for document KIE by shifting from general-purpose LLM scaling to domain-specific adaptation tailored for concrete application scenarios. The contributions of this work are summarised as follows: **First**, **we shift the paradigm from general scaling to domain-focused scaling for specialized document KIE tasks.** We develop a synthetic data generation workflow that integrates OCR, PDF parsing, multi-task inquiry generation, and quality-verification modules. By applying the scaling law to the target domain, this pipeline produces large volumes of high-quality synthetic annotations that accurately capture both the structure and content of complex, domain-specific documents. **Second**, **we introduce the concept of a "warmer"—a lightweight discriminative retriever designed to adapt domain-specific knowledge and supply grounded multimodal cues to generative MLLMs.** By training only this efficient retriever rather than fine-tuning large MLLMs, the framework offers substantial computational savings and reduces dependence on large manually annotated datasets through synthetic data supervision. In addition, through iterative knowledge exchanging between warmer and MLLM, outputs from the discriminative and generative models are iteratively refined, yielding more context-aware, stable, and accurate zero-shot responses. **Third**, **we introduce domain adaptation strategies to train the joint-grained warmer, enabling it to bridge the gap between MLLM knowledge and target-domain documents.** Structural adaptation improves semantic–spatial understanding, while semantic adaptation strengthens user intent and content comprehension, both essential for strong zero-shot performance. After domain-adaptive tuning, the warmer can "warm up" domain knowledge for MLLM-based document KIE, enabling more accurate, contextually grounded inference. **Finally**, by integrating these components, SynDoc delivers a scalable and robust framework for domain-specific key information extraction, validated on three in-domain datasets and further tested for cross-domain generalization.

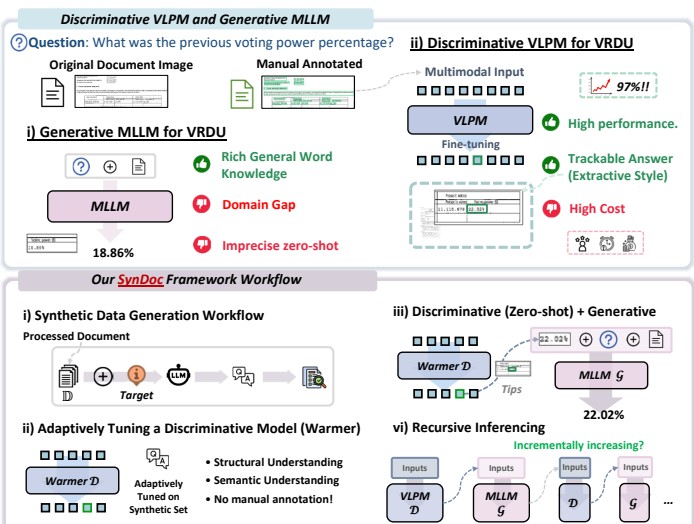

Figure 1: Comparing SynDoc with discriminative and generative document KIE frameworks.

## 2 RELATED WORK

**Curated and synthetic data for VRDU**   Heuristic approaches (Watanabe et al., 1995; Seki et al., 2007) and statistical learning methods (Oliveira & Viana, 2017) have demonstrated strong performance in domain-specific document understanding. However, these methods rely heavily on expert effort, which limits their cross-domain adaptability. To address this limitation, recent advances employ self-supervised learning on large-scale, unannotated, and multi-source document collections (Huang et al., 2022; Tang et al., 2023; Lyu et al., 2024; Wang et al., 2022a; Hong et al., 2022; Harley et al., 2015), thereby improving generalizability and multimodal comprehension for broader VRDU tasks. When fine-tuned on curated datasets, these frameworks can achieve state-of-the-art performance in specific VRDU tasks. Yet, constructing such high-quality datasets is resource-intensive, hindering scalability and applicability to new document types (Jaume et al., 2019; Park et al., 2019; Ding et al., 2023b). To mitigate this bottleneck, recent research (Ding et al., 2024b) has explored using

LLMs/MLLMs to generate synthetic datasets with well-designed prompts and human verification. Other studies create large-scale synthetic datasets for self-supervised pretraining (Hu et al., 2024a; Feng et al., 2024) or instruct-tuning (Wang et al., 2025; Tang et al., 2024; Zhang et al., 2025) to enhance multimodal document understanding. For example, Ding et al. (2024a) pretrain VRDU models with synthetic QA pairs, followed by semi-supervised refinement, achieving performance comparable to full supervision. However, optimizing the quality of synthetic dataset generation and integrating SoTA MLLMs into real-world VRDU applications remains underexplored.

**VRDU frameworks** Self-supervised frameworks employ diverse pre-training tasks to enhance multimodal learning and achieve strong performance on downstream tasks when fine-tuned with curated datasets (Wang et al., 2022b; Appalaraju et al., 2023). However, most discriminative models rely heavily on off-the-shelf OCR tools, including the LayoutLM-series, making their extractive predictions vulnerable to errors and to propagation from both models and OCR systems (Huang et al., 2022; Xu et al., 2021). To address this vulnerability, end-to-end OCR-free frameworks have been developed to bypass OCR dependencies (Kim et al., 2022; Abramovich et al., 2024; Lyu et al., 2024). Nonetheless, these OCR-free models often operate with comparatively smaller parameters and limited training resources, which constrains their world knowledge and reduces generalization capacity without substantial annotations. By contrast, LLMs/MLLMs (OpenAI, 2024; Team et al., 2024; Laurençon et al., 2024; Yang et al., 2025) benefit from scaling laws and extensive training to capture broad world knowledge and support zero-shot or few-shot learning in VRD tasks (He et al., 2023). Despite these advantages, challenges such as hallucinations and limited domain-specific knowledge continue to undermine their reliability. To bridge this gap, our SynDoc introduces an adaptively tuned discriminative "warmer" that provides domain-specific knowledge to a generative MLLM. This integration enables recursive refinement of the inference process, combining domain-aware information and broad world knowledge to enhance the accuracy and reliability of key information extraction in VRDs.

## 3 METHODS

### 3.1 OVERVIEW OF SYNDOC

Let $\mathbb{D}$ be a document collection within a *specific domain*. We introduce SynDoc, a hybrid framework designed to answer a user-provided natural language query $q$ concerning a specific document $d \in \mathbb{D}$. SynDoc integrates a discriminative model $\mathcal{D}$ and a generative model $\mathcal{G}$ to address $q$ in complementary extractive and abstractive manners, respectively. $\mathcal{D}$, termed **warmer**, employs pretrained backbones to capture target-domain knowledge, while $\mathcal{G}$ employs state-of-the-art LLM/MLLMs, guided by carefully designed prompts $P$, to generate answers in zero-shot scenarios.

To enable this workflow, we first construct the **synthetic datasets** (Figure 2). This process begins

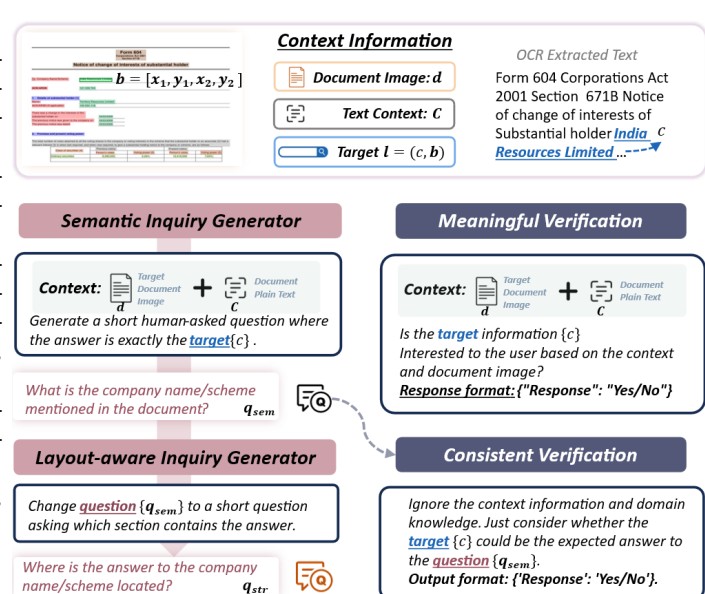

Figure 2: Workflow of the Synthetic Data Generator.

with the extraction of structural information using off-the-shelf tools (e.g., OCR or PDF parsers (Cui et al., 2025)). Next, domain-specific synthetic queries are generated with MLLMs, focusing separately on structural and content understanding. $\mathcal{D}$ incorporates multimodal representations, including textual, visual, layout, and structural features, alongside predictions from the MLLM.

During inference, the outputs from $\mathcal{D}$ and $\mathcal{G}$ undergo **iterative cross-feeding** until stable predictions are reached.

The following subsections detail the four key modules of SynDoc: (1) Synthetic Data Generator, (2) Discriminative Warmer Architecture, (3) Adaptive Instruction Tuning, and (4) Recursive Inference.

## 3.2 SYNTHETIC DATA GENERATOR

**VRD Structure Parsing**   We use off-the-shelf tools to extract both the **text content** and **layout structure** of a target document collection (Figure 2). For **document images**, we use a vision-based OCR system to extract text-line entities $L$. Each $l = (b, c) \in L$ includes a bounding box $b$ and text content $c = \{\tau_i\}$, where each $\tau$ is a textual token. Bounding boxes are represented by coordinates $(x_{min}, y_{min}, x_{max}, y_{max})$. For **text-embedded PDF files**, we employ PDF parsing tools to acquire text-line or semantic entity sets $L$ (e.g., paragraph, list, section), which provide more accurate structural information than OCR alone.

**MLLM-driven Inquiry Generation**   For $\mathcal{D}$ to capture target-domain knowledge, we propose an MLLM-driven workflow with two modules (Figure 2). (1) **Multi-Task Inquiry Generation** produces diverse queries to instruction-tune $\mathcal{D}$ to enhance its structural and semantic understanding of the domain. Specifically, a set of text lines is randomly sampled and fed to an LLM to generate two types of QA pairs. First, *Semantic* QA pairs guide $\mathcal{D}$ to extract target information from a document. Given the content of a target entity along with its document and context information, the MLLM generates pairs $(q_{sem}, c)$, where $c$ is the answer to the generated question $q_{sem}$. Second, *Spatial-aware* QA pairs help $\mathcal{D}$ capture both semantic and spatial correlations. Here, each $q_{sem}$ is transformed into a spatially-aware question $q_{spt}$ by identifying the document region (e.g., top-left, top-middle, top-right) where the target information $c$ is located. (2) **Multi-Aspect Quality Verification** is designed to filter out low-quality or noisy questions by assessing factors such as meaningfulness and question-answer consistency. This step ensures that synthetic QA pairs are both relevant and reliable for instruction-tuning $\mathcal{D}$. First, the process determines whether $c$ is relevant to the end user (e.g., "Is $c$ meaningful or interesting to the end user?"). Next, it verifies that $c$ adequately addresses $q_{sem}$ (e.g., "Could $c$ reasonably serve as an answer to $q_{sem}$?").

## 3.3 WARMER ARCHITECTURE

The **Warmer** ($\mathcal{D}$) utilizes a vision-language pre-trained model (VLPM) as its backbone, optimized for discriminative answer extraction through adaptively tuning on synthetic datasets (Figure 3). The adopted VLPM is pretrained on layout-aware tasks and fine-tuned on well-annotated datasets, exhibiting strong performance on targeted VRDU tasks. To address zero-shot scenarios, the warmer architecture leverages the VLPM backbone to enable $\mathcal{D}$ to learn multi-aspect, domain-aware knowledge from synthetic datasets. In the following, we first introduce the initial feature representation used by $\mathcal{D}$ and then describe the detailed warmer architecture.

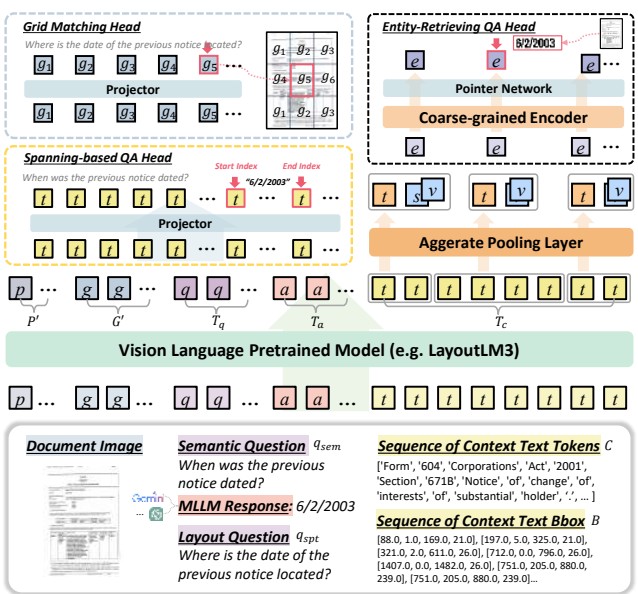

Figure 3: Architecture of the discriminative Warmer.

**Initial Feature Representation**   For a synthetically acquired entity set $L$ from a document $d \in \mathbb{D}$, following Ding et al. (2024b), a vision pretrained model extracts visual representations $v$ and a text pretrained model extracts textual sentence embedding $s$ of any textline $l \in L$. A textual sequence $C = \{\tau_i\}_{i=1}^n$ represent textual context of target

document $d$, combined with the projected bounding box $B = \{b_i\}_{i=1}^n$ and concatenated with document image patches $P$. For each semantic query $q_{sem}$, the MLLM-generated answer $a$ can provide additional guidance for localization. Finally, grid embeddings $G = \{g_i\}_{i=1}^{j \times k}$ result from resizing and flattening the document image into a $j \times k$ grid, capturing global spatial layout information.

**Detailed Architecture** $\mathcal{D}$ processes the input sequence ($q$, $a$, $C$, $P$ and $B$). These inputs are fed into a VLPM backbone $\mathcal{E}_w$ to derive embedded feature representations:

$$(P', G', T_q, T_a, T_c) = \mathcal{E}_w(P, G, q, a, C + B). \tag{1}$$

Here, $T_q$, $T_a$, and $T_c$ denote the encoded textual features for the query, answer, and context, respectively, while $P'$ and $G'$ represent the encoded patch and grid features of the document image.

For each textline $l = (b, c) \in L$ extracted using parsing tools, a pooling layer aggregates the token features to obtain the entity-level representation $e$,

$$e = \text{Pooling}\big(\{\, \mathcal{E}_w(t_j) \mid t_j = \tau_j + b, \ \tau_j \in c \,\}\big) \oplus v \oplus s. \tag{2}$$

The enhanced entity features $E = \{e_l \mid l \in L\}$ are processed by an **Entity-Retrieval Head**, which includes a coarse-grained transformer encoder to improve entity-level contextual understanding and a pointer network to predict the final entity index (Ding et al., 2024b). Additionally, a fine-grained **Span-based QA Head** gets the encoded document text feature $T_c$ to predict the start and end indices of the answer span for input query $q$. To enhance structural understanding within the target domain, a **Grid Matching Head** gets encoded grid embedding $G'$ to predict the grid index of spatial-aware queries. Each of these heads may be trained at different stages to enable the Warmer to capture multi-aspect, domain-specific knowledge.

### 3.4 ADAPTIVELY WARMER TUNING

Step-by-step training enables the warmer $\mathcal{D}$ to effectively adapt to the target domain, starting with **structural adaptation** to enhance the domain-specific layout understanding, followed by **semantic adaptation** to locate target information based on the input query.

**Structural Adaptation** enhances both semantic and spatial comprehension by guiding $\mathcal{D}$ to identify the most relevant document grid $g' \in G'$ for a given structural query $q_{str}$. For example, given the query "*Where is the date of the previous notice located?*", $\mathcal{D}$ predicts the grid $g_5$ that contains the answer (Figure 3). A pointer network computes logits for each candidate grid (Ding et al., 2024b), and the probabilities are obtained using the softmax function. The structure adaptation loss is defined using cross-entropy:

$$\mathcal{L}_{str} = -\sum\nolimits_{g' \in G'} y_{g'} \log \hat{y}_{g'}, \tag{3}$$

where $y_{g'}$ represents the ground truth grid. This process ensures that $\mathcal{D}$ learns to associate structural queries with the correct document regions, improving retrieval accuracy and layout-aware reasoning.

**Semantic Adaptation** enables $\mathcal{D}$ to pretrain on a synthetic semantic QA set $P$, allowing it to understand document image $I_d$ and $q_{sem}$ for zero-shot extractive QA. The model employs two extractive QA heads. (1) *Fine-grained Span-based QA Head* predicts the start and end token indices using a linear projector. The cross-entropy loss is $\mathcal{L}_{fg} = -\sum_{t \in \mathcal{E}_w(c)} y_t^{\text{start}} \log \hat{y}_t^{\text{start}} + y_t^{\text{end}} \log \hat{y}_t^{\text{end}}$ where $y_t^{\text{start}}$ and $y_t^{\text{end}}$ denote the ground truth indices, and $\hat{y}_t^{\text{start}}$ and $\hat{y}_t^{\text{end}}$ are the predicted probabilities after softmax. (2) *Coarse-grained Entity-Retrieving Head* retrieves target entities based on entity logits and is optimized with a cross-entropy loss: $\mathcal{L}_{cg} = -\sum_{e \in E} y_e \log \hat{y}_e$ where $y_e$ represents the ground truth distribution over the entity set $E$, and $\hat{y}_e$ is the predicted softmax probability. The final optimization objective combines both losses as:

$$\mathcal{L} = \lambda_{fg} \mathcal{L}_{fg} + \lambda_{cg} \mathcal{L}_{cg}, \tag{4}$$

where $\lambda_{fg}$ and $\lambda_{cg}$ balance the contributions of the fine-grained and coarse-grained QA losses. During semantic adaptation, different synthetic subsets may be selected based on *Multi-Aspect Quality Verification*, leading to potential variations in model performance, as described in Section 5.2.

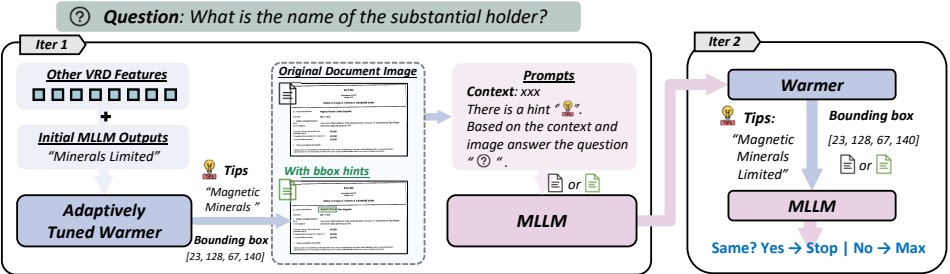

Figure 4: Recursive inference framework for zero-shot QA on VRDs. Given a query, the initial MLLM-generated answer is iteratively refined using retrieved entity hints ($L_\mathcal{D}$) retrieved by the adaptively tuned Warmer. Bounding-box cues and other VRD features guide the MLLM toward more accurate and context-aware answers in subsequent iterations.

## 3.5 RECURSIVELY INFERENCING

We apply a **recursively inferencing strategy** to jointly leverage $\mathcal{D}$ and $\mathcal{G}$ for zero-shot question answering on VRDs (Figure 4). The top-$k$ entities retrieved by $\mathcal{D}$, denoted $L_\mathcal{D}$, provide domain-specific guidance to enhance MLLM responses. Initially, given the prompt $(I_d, C, q_{sem}) \to \Pi$, $\mathcal{G}$ generates an answer $A_\mathcal{G}$. During the $t$-th recursive step, $\mathcal{D}$ updates its retrieval based on the previous $A_\mathcal{G}^{(t)}$, which in turn informs an updated prompt:

$$L_\mathcal{D}^{(t+1)} = \mathcal{D}(A_\mathcal{G}^{(t)}), \quad \Pi^{(t+1)} = \text{UpdatePrompt}(\Pi^{(t)}, L_\mathcal{D}^{(t+1)}), \quad A_\mathcal{G}^{(t+1)} = \mathcal{G}(\Pi^{(t+1)}). \quad (5)$$

This iterative process allows $\mathcal{G}$ to progressively incorporate domain-specific knowledge, improving the accuracy and reliability of its answers. Recursive inference continues until a maximum recursion depth is reached or the generated answer stabilizes. Through this approach, both extractive and generative outputs are iteratively refined.

## 4 EXPERIMENTAL SETTINGS

### 4.1 DATASETS

We evaluated SynDoc on four domain-specific document KIE datasets: FormNLU (financial forms) (Ding et al., 2023a), CORD (receipts) (Park et al., 2019), Ephoie (exam papers) (Wang et al., 2021), and FUNSD (Jaume et al., 2019) (multi-domains) (Appendix A.1 for more details). Form-NLU is further divided into Printed (F-P) and Handwritten (F-H) subsets. For each *test set*, document images were processed using the *Synthetic Data Generation* module to produce synthetic structural annotations and QA pairs with quality verification. During inference, the original QA pairs or key-value/question pairs are used to evaluate performance.

For the FUNSD and CORD datasets, we utilized the processed test sets from Luo et al. (2024). For Form-NLU and Ephoie, key-value pairs were converted into QA pairs for inference. Following prior works (Mathew et al., 2021; Luo et al., 2024), we adopted the Averaged Normalized Levenshtein Similarity (ANLS) as the primary **evaluation metric**.

### 4.2 BASELINES AND IMPLEMENTATION DETAILS

We evaluated SynDoc against a range of SoTA baselines, including open-source models (Qwen2-VL (Wang et al., 2024), Idefics2 (Laurençon et al., 2024), InternVL2 (Chen et al., 2024)) and proprietary systems (GPT-4o (OpenAI, 2024), Gemini 1.5 (Team et al., 2024)). These models were selected for their demonstrated effectiveness in document-centric tasks, including comprehension, retrieval, and question answering. For consistency, all MLLMs were assessed under default inference settings in the HuggingFace environment using up to $2\times$ A100 80G GPUs. Warmer tuning employed a batch size of 16, a learning rate of $2e-5$, and the AdamW optimizer.

## 5 RESULTS AND DISCUSSION

### 5.1 MAIN RESULTS

Table 1 shows that proprietary models generally outperform their open-source counterparts, with the gap most pronounced in complex scenarios (e.g., F-H and Ephoie). Among open-source MLLMs of similar size, Qwen2-VL achieves the highest performance, likely due to its extensive multimodal training data and advanced OCR capabilities. Intern-VL2 also performs consistently well across all datasets, whereas Idefics2 struggles with structurally complex documents, particularly in Ephoie.

Since Gemini shows superior performance across most benchmark datasets compared to GPT-4o, we present the results of the Gemini-based Syn-

Table 1: Zero-shot MLLM results. The last row shows the best configuration with bounding boxes.

| Model | F-P | F-H | CORD | Ephoie | FUNSD |
|---|---|---|---|---|---|
| Idefics2 | 57.54 | 33.31 | 54.45 | 15.22 | 62.11 |
| InternVL2 | 66.56 | 45.47 | 66.84 | 68.92 | 74.95 |
| Qwen2-VL | 78.05 | 43.65 | 77.86 | 70.36 | 79.12 |
| GPT-4o | 76.16 | 56.49 | 79.05 | 79.40 | 80.05 |
| Gemini | 76.09 | 66.86 | 84.35 | 81.82 | 83.56 |
| SynDoc (Gemini) | | | | | |
| Top-1 | 80.29 | 67.73 | 85.19 | 81.80 | 82.77 |
| Top-$K$ | 81.60 | 66.90 | 83.57 | 81.33 | 82.12 |
| Top-1 R | 80.29 | 67.73 | 85.19 | **82.15** | 83.02 |
| Top-$K$ R | **81.91** | 68.09 | 84.57 | 81.58 | 82.40 |
| w/bbox | 80.93 | **68.13** | **85.40** | 82.08 | **83.87** |

Doc framework. Overall, incorporating adaptively tuned warmer knowledge into MLLMs enhances performance on domain-specific datasets; however, it can introduce noise in cross-domain benchmarks such as FUNSD. The results also suggest that employing top-$K$ candidate hints or applying recursive inference (top-$K$ R) substantially improves MLLM performance in zero-shot scenarios.

### 5.2 WARMER PERFORMANCE ANALYSIS

Here, we evaluated the effectiveness of the *Synthetic Data Generation* workflow and *Warmer*'s ability to capture domain-specific knowledge in three aspects: adaptive tuning strategies, top-$k$ entity retrieval variations, and different pretrained backbones. These experiments aim to determine whether the proposed methods optimize Warmer's performance and enhance information extraction in zero-shot scenarios, thereby providing strong support for downstream MLLM inference. Notably, Table 2 shows that, without tuning on synthetically generated data, the pretrained Warmer failed to retrieve any meaningful information from the target document context.

**Adaptive Tuning Strategies** We first evaluated the Warmer's performance under different adaptive tuning methods in adaptive tuning sets, prior MLLM outputs, and structural-adaptive tuning.

Table 2: Results under various Warmer adaptive tuning settings. Adapt denotes 4 configurations: (1) full synthetic set, (2) meaningfulness verification subset, (3) consistency verification subset, and (4) both subsets. St - structure adaptation. Prior - prior MLLM outputs.

| Adapt | St | Prior | F-P | F-H | CORD | **Ephoie** | FUNSD |
|---|---|---|---|---|---|---|---|
| N/A | ✗ | ✗ | 0 | 0 | 0 | 0 | 0 |
| 1 | ✗ | ✗ | 31.39 | 18.18 | 41.48 | 19.23 | 44.37 |
| 2 | ✗ | ✗ | 42.56 | 16.41 | 46.71 | 20.64 | 48.66 |
| 3 | ✗ | ✗ | 33.87 | 14.61 | 41.16 | 22.74 | 42.77 |
| 4 | ✗ | ✗ | 44.23 | 12.23 | 50.44 | 23.78 | 44.67 |
| 1 | ✗ | ✓ | 59.26 | 30.67 | 65.60 | 22.94 | 56.83 |
| 2 | ✗ | ✓ | 65.67 | 31.63 | 66.37 | 22.06 | 57.77 |
| 3 | ✗ | ✓ | 64.68 | 27.85 | 65.9 | 25.48 | 57.43 |
| 4 | ✗ | ✓ | 65.75 | 29.31 | 65.08 | 24.76 | 59.86 |
| 1 | ✓ | ✓ | 62.67 | 30.25 | 66.21 | 24.12 | 58.08 |
| 2 | ✓ | ✓ | 66.03 | **31.64** | **67.26** | 24.13 | 58.05 |
| 3 | ✓ | ✓ | 65.2 | 28.83 | 63.94 | 25.29 | 61.01 |
| 4 | ✓ | ✓ | **66.19** | 28.29 | 66.25 | **27.16** | **61.24** |

i) *Effects of adaptive tuning sets.* Table 2 shows that both verification methods improve performance and enhance domain adaptation. In particular, meaningfulness verification consistently boosts performance, whereas consistency verification occasionally has a negative impact, likely because OCR errors lead to inaccurate MLLM justifications.

ii) *Impact of prior MLLM outputs.* Table 2 shows that incorporating MLLM outputs as Warmer input further improves its ability to locate relevant information. For instance, ANLS increased from 41.48 to 66.37 on CORD, 44.37 to 61.24 on FUNSD, and 19.23 to 27.16 on Ephoie. While the synthetic dataset remains the primary source of the Warmer's knowledge, providing MLLM inputs helps it efficiently locate target answers.

iii) *Structural Adaption Tuning.* This mechanism enhances the Warmer's comprehension of layout and semantic correlations within a specific domain. Table 2 demonstrates consistent improvements across datasets: CORD increasing from 50.44 to 67.26, FUNSD from 44.67 to 61.24, and Ephoie

from 22.74 to 27.16, when comparing the best results with and without structural adaptation. These result indicates that self-supervised structural adaptation effectively warms up the Warmer, enabling richer structural and semantic representations and enhancing subsequent semantic adaptation.

**Top-$K$ Retrieved Entity Performance** While the Warmer's maximum likelihood prediction may not always yield the most relevant result, leveraging Top-$K$ likelihood predictions can enhance the MLLM's ability to locate the correct answer (Section 5.1).

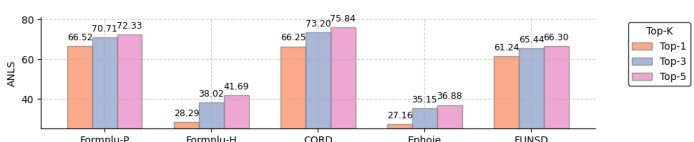

Figure 5: Top-$K$ retrieved entity performance using LayoutLMv3.

We compared Top-1, Top-3, and Top-5 retrieved entities, selecting the entity with the highest ANLS when multiple candidates are available.

Figure 5 shows that the Top-3 predictions significantly improve the retrieval of relevant information compared to Top-1. However, the gain from Top-3 to Top-5 is marginal. Notably, the improvement from Top-1 to Top-3 is especially pronounced for datasets with lower OCR accuracy, indicating the benefit of broader retrieval in error-prone scenarios.

**Various Warmer Backbones** We selected three commonly used pretrained backbones to assess their effectiveness: the text-only RoBERTa (Liu, 2019), the text-and-layout-aware LiLT (Wang et al., 2022a), and the text, layout, and vision-aware LayoutLMv3 (Huang et al., 2022). Table 3 shows that multimodal frameworks tend

Table 3: Comparison of results under different Warmer backbone configurations.

| Model | F-P | F-H | CORD | Ephoie | FUNSD |
|---|---|---|---|---|---|
| Roberta | 64.18 | 23.85 | **70.40** | 31.57 | 59.44 |
| LiLT | 63.82 | 30.89 | 67.87 | **31.97** | **60.94** |
| LayoutLMv3 | **65.75** | **31.63** | 66.37 | 25.48 | 59.86 |

to outperform the monomodal RoBERTa, particularly when OCR errors impact the input text. However, LayoutLMv3 exhibits weaker feature representations, significantly underperforming LiLT and RoBERTa despite all three using the same xlm-RoBERTa-base checkpoints. Interestingly, there are instances where the monomodal RoBERTa outperforms multimodal backbones, indicating that multimodal architectures do not always guarantee superior performance or enhanced domain-specific knowledge extraction.

## 5.3 ABLATION STUDIES

Here, we assessed the effectiveness of the zero-shot trained Warmer for enhancing MLLM inference and explored the impact of the recursive inference mechanism across various MLLMs.

**Performance on Various MLLMs** Table 4 reports the results for two high-performing open-source models (InternVL and QWenVL) alongside the best-performing proprietary model (Gemini). Across all models and datasets, incorporating Warmer outputs

Table 4: Comparison of Warmer to Generative Models.

| | F-P | | F-H | | CORD | | Ephoie | |
|---|---|---|---|---|---|---|---|---|
| Model | Vani. | Ours | Vani. | Ours | Vani. | Ours | Vani. | Ours |
| InternVL | 66.56 | ↑ 68.09 | 45.47 | ↑ 46.81 | 66.84 | ↑ 68.80 | 68.92 | ↑ 70.29 |
| QWenVL | 78.05 | ↓ 77.27 | 43.65 | ↑ 44.43 | 77.86 | ↑ 78.44 | 70.36 | ↑ 75.03 |
| Gemini | 76.09 | ↑ 81.91 | 66.86 | ↑ 68.02 | 84.35 | ↑ 85.19 | 81.82 | ↑ 82.15 |

consistently improves performance, suggesting that the Warmer successfully captures task-specific patterns that generic MLLMs might overlook.

**Recursive Warmer Performance** Table 5 shows that recursive inference enhances the performance of both discriminative Warmer and generative MLLM. Notably, the FormNLU dataset exhibits substantial gains, with scores rising from 66.19 to 73.76 on the printed subset and from 31.64 to 39.15 on the handwritten subset.

Table 5: Impact of iteration count on Warmer and Gemini.

| | F-P | | F-H | | CORD | | Ephoie | |
|---|---|---|---|---|---|---|---|---|
| | Warmer | Gemini | Warmer | Gemini | Warmer | Gemini | Warmer | Gemini |
| N/A | 66.19 | 76.09 | 31.64 | 66.86 | 67.26 | 84.35 | 27.16 | 81.82 |
| 1 | 73.57 | **80.29** | 38.11 | **67.73** | 63.37 | **85.19** | **27.98** | 81.80 |
| 2 | **73.76** | 80.17 | 38.79 | 67.60 | 64.15 | 84.67 | 25.94 | 81.91 |
| 3 | 73.72 | 80.15 | **39.15** | 67.32 | 64.32 | 84.65 | 26.03 | 81.71 |
| 4 | 73.76 | 79.88 | 38.84 | 67.63 | 64.32 | 84.39 | 25.94 | **82.15** |
| 5 | 73.60 | 80.06 | 38.92 | 67.63 | 64.04 | 84.40 | 26.12 | 81.86 |

Interestingly, the peak performance of Warmer and MLLM does not always coincide at the same iteration. This suggests that, while Warmer steadily improves retrieval quality, Gemini might require

additional iterations to fully leverage these improvements during its integration and reasoning process.

**Effectiveness of Top-$K$ Candidates**
Figure 6 shows that providing top-$K$ candidates from the Warmer can increase the likelihood of integrating relevant extracted information into MLLMs and improve performance. For instance, in FormNLU, retrieving additional information from the Warmer help guide Gemini to focus on the relevant context, thereby enhancing its performance. However, this approach also carries the risk of incorporating noise into the prompt, which may negatively impact the generative model's performance. This effect is particularly notable in InternVL2 and QWenVL2 when applied to OCR-challenging datasets, such as F-H and Ephoie.

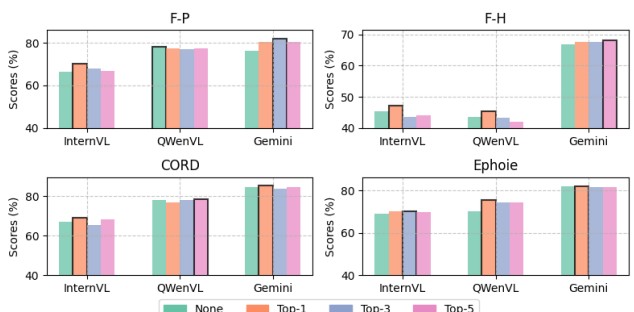

Figure 6: Result comparison by feeding Top-$K$ Warmer-Retrieved Candidates into MLLM.

## 6  CASE STUDY

To further illustrate the effectiveness of SynDoc, Figure 7 presents several examples where initial MLLM predictions are refined using SynDoc. Additional examples are provided in Appendix F. In **Q1**, a question regarding the present voting count initially yields an incorrect answer of "15,41", which is subsequently corrected to "27,210" with the aid of the Warmer. This example highlights how the Warmer injects domain-specific knowledge, mitigating hallucinations and reducing the imprecision of MLLM predictions. Relying solely on the Top-1 retrieved answer from the Warmer, however, may not always capture the most relevant context for accurate answering. As demon-

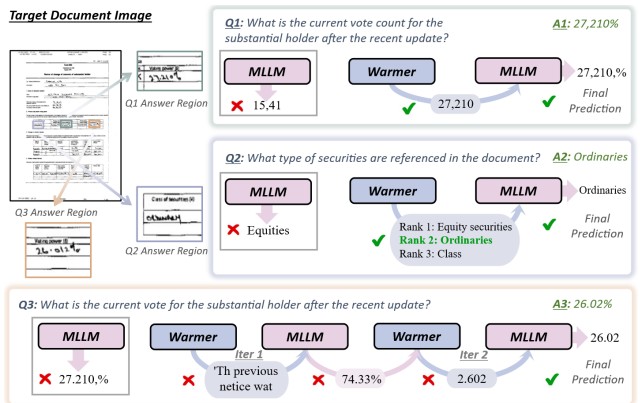

Figure 7: Qualitative Case Studies for Q1) Effectiveness of Warmer Retrieving; Q2) Demonstrating the Top-K candidates; 3) Effectiveness of iterative Inferencing.

strated in **Q2**, providing Top-3 entities allows the model to leverage both the Warmer's domain-specific knowledge and the MLLM's general world knowledge, thereby refining the final prediction. Finally, **Q3** showcases the iterative inference mechanism. Here, the Warmer and MLLM improve each other's outputs, leading to an almost correct prediction. Notably, even when the Warmer provides the ideal hints in the final iteration, OCR errors may persist. However, the MLLM compensates for these issues by leveraging its extensive general knowledge to make accurate predictions.

## 7  CONCLUSION

In this paper, we introduced SynDoc, a novel VRDU framework that seamlessly integrates discriminative VLPMs with generative MLLMs to advance domain-specific VRDU performance, particularly in zero-shot settings. Our extensive experiments show that the proposed *Synthetic Data Generator* and *Adaptive Warmer Tuning* enable the discriminative Warmer to efficiently acquire domain knowledge and, together with recursive inference, drive continual performance gains for both the warmer and the MLLM. While SynDoc exhibits strong results across multiple domain-specific datasets, further work may be needed to enhance its generalizability and robustness in cross-domain applications.

ETHICS STATEMENT

This work adheres to the ICLR Code of Ethics. In this study, no human subjects or animal experimentation were involved. All datasets used were sourced in compliance with relevant usage guidelines, ensuring no privacy violations. We have taken care to avoid any biases or discriminatory outcomes in our research process. No personally identifiable information was used, and no experiments were conducted that could raise privacy or security concerns. We are committed to maintaining transparency and integrity throughout the research process.

REPRODUCIBILITY STATEMENT

We have made every effort to ensure that the results presented in this paper are reproducible. All code and models will be made publicly available upon acceptance to enable verification and replication. As detailed in Section 4.2, our experimental setup is fully specified, including the training procedure, model architecture, and computational environment.

Additionally, all datasets used are publicly accessible, ensuring consistency in our evaluation. We believe these measures will enable other researchers to reproduce our work and further advance the field.

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

# A   DETAILED DATASET INFORMATION

## A.1   DATASET DESCRIPTION

Table S1 summarizes the datasets used in this study.

**Form-NLU** is introduced for financial-domain form layout and content understanding, focusing on single-template, multi-format forms across digital, printed, and handwritten variations (Ding et al., 2023a). This dataset specifically addresses KIE tasks, which involve extracting 12 types of key information from more challenging printed and handwritten documents. Examples of these key information fields include "*Substantial Holder Name*", "*Previous Persons' Votes*", and others.

**CORD** is proposed for receipt understanding with diverse receipt templates (Park et al., 2019). This dataset focuses on the KIE sub-task of extracting fine-grained key information from scanned receipts, such as "*store name*" and "*item quantity*".

**Ephoie** is a dataset designed to understand scanned Chinese exam paper headers (Wang et al., 2021). The collected exam papers have diverse templates and handwritten information. This dataset focuses on the KIE sub-task to extract information from these exam papers, such as "*Score*," "*School*," and "*Student Name*."

**FUNSD** is a dataset for form understanding, comprising scanned form images from diverse sources with varying templates (Jaume et al., 2019). Each form contains predefined key-value pairs categorized as "*Question*" and "*Answer*" in the metadata. This dataset is used to assess the proposed framework's ability to handle cross-domain scenarios.

Table S1: Dataset statistics across different datasets, including the size of the original test and the synthetic dataset.

| Domain | Category | # Doc | # QA | Set 1 | Set 2 | Set 3 | Set 4 |
|---|---|---|---|---|---|---|---|
| FormNLU-P | Financial Form | 50 | 596 | 1937 | 1137 | 1073 | 676 |
| FormNLU-H | Financial Form | 50 | 597 | 1998 | 621 | 815 | 302 |
| CORD | Receipt | 100 | 156 | 1644 | 1535 | 988 | 968 |
| EPHOIE | Exam Paper | 311 | 928 | 2488 | 1746 | 1553 | 1159 |
| FUNSD | Cross-domain | 50 | 467 | 2036 | 1905 | 1088 | 1022 |

# B  DETAILED MODEL INFORMATION

Table S2 summarizes baseline models for visual-rich document understanding.

Table S2: Baseline Models for Visual-rich Document Understanding.

| Model | Params | Modality | Training Data | Status |
|---|---|---|---|---|
| RoBERTa | 125M | Text | Web, Books | Open |
| LiLT | 131M | Text+Layout | IIT-CDIP | Open |
| LayoutLMv3 | 133M | Text+Layout+Vision | IIT-CDIP | Open |
| GPT-4o | ∼200B | Text+Vision+Audio | Web+Images+Audio | Closed |
| Gemini 1.5 | 175B | Text+Vision+Audio | Web+Multimodal | Closed |
| InternVL2 | 8B | Text+Vision | Documents, Medical | Open |
| QwenVL2 | 72B | Text+Vision+Video | Web, OCR, Video | Open |
| Idefics2 | 8B | Text+Vision | Web, Documents | Open |

## B.1  WARMER VARIANTS DETAILS

**RoBERTa** is a self-supervised text-only language model trained on a large corpus (Liu, 2019), including BookCorpus, English Wikipedia, CommonCrawl News, OpenWebText, and Stories datasets. RoBERTa removes the next-sentence prediction (NSP) objective and uses dynamic masking, larger batch sizes, and longer sequences.

**LiLT** (Language-independent Layout Transformer) extends pretrained text encoders with a lightweight layout encoder (Wang et al., 2022a). It is pretrained on the IIT-CDIP scanned document corpus. LiLT features a dual-stream architecture to separately encode text and layout (bounding box) information, with Bi-directional Attention Complementation (BiACM) to enhance cross-modal alignment.

**LayoutLMv3** is a multimodal Transformer that jointly encodes text, layout, and image information (Huang et al., 2022). It is pretrained on the IIT-CDIP corpus and synthetic document data, using masked language modeling (MLM), masked image modeling (MIM), and word-patch alignment (WPA) tasks.

## B.2  LARGE VISION-LANGUAGE MODELS DETAILS

### B.2.1  CLOSE SOURCE MODELS

**GPT-4o** is a multimodal model capable of processing text, images, and audio, with an estimated parameter count of hundreds of billions to 1 trillion (OpenAI, 2024). Trained on web-scale text, images, and audio, GPT-4o features native multimodal reasoning, multilingual support, and high-speed inference.

**Gemini 1.5 Pro** is a mid-size multimodal model with a Mixture-of-Experts (MoE) architecture, trained on a vast multimodal corpus with a focus on long-context tasks up to 1 million tokens (Team et al., 2024): .

### B.2.2  OPEN SOURCE MODELS

**InternVL2** combines a vision Transformer and a language model (Chen et al., 2024). It is pretrained on 5M curated multimodal samples, including documents, forms, scientific charts, and medical images. InternVL2 ranges from 1B to 108B parameters and is pretrained on curated multimodal data, including documents, forms, scientific charts, and medical images. It achieves competitive results on specific document-centric tasks, such as DocVQA.

**QwenVL2** is trained on 1.4T tokens, including image-text pairs, OCR data, video, and interleaved documents (Wang et al., 2024). With innovations like Naive Dynamic Resolution and Multimodal RoPE, QwenVL2 achieves competitive performance on multimodal benchmarks, establishing itself as a leading open-source option.

**Idefics2** combines a Mistral-7B language model with a SigLIP vision encoder (Laurençon et al., 2024). Trained on interleaved web documents, captions, OCR data, and diagram-text mappings, it

supports arbitrary sequences of text and images. Despite its smaller size, it achieves performance comparable to that of 30B+ models.

## C  DETAILED PROMPTS

We list all the prompts used in this paper for synthetic data generation in Table S3 and MLLM zero-shot testing in Table S4.

Table S3: Synthetic Data Generator Prompt Example

| Module | Prompt Description | Prompt Template |
|---|---|---|
| User-Input Verification | Checks whether the target information was entered by the user or is part of the form template. | Based on the provided Context {} from the target form and the form image itself, check if the target information itself (do not consider the context) {} was entered by the form user (not part of the form template). Only output "Yes" if the {} is exactly provided by the user, not from the form template, and do not consider context information. The response should follow the format below: "Response": "Yes/No" |
| Semantic Question Generation | Generates a short human-asked question where the answer exactly matches the target. | Based on the above context {} and target document image, generate a human-asked SHORT question (output question only) of which answer is exactly the same as {} |
| Answer Verification | Verifies whether the given target could be the expected answer to the given question. | Ignore the context information and domain knowledge (e.g., FAX NUMBER). Just consider whether {} could be the expected answer to the question {}. Output format: {'Response': 'Yes/No', 'Explanation': 'xxx'}. |
| Layout-Aware Question Reformulation | Reformulates a question into a short question about the location of the answer in the document. | Change the question {} to a very short question about finding the position of the answer from the input document image. For example, where is the answer to xx located? |

Table S4: Summary of Inference Prompt Functions and Their Templates

| Module | Prompt Description | Prompt Template |
|---|---|---|
| Text-Image QA without Tips | Generates a response to a question based on an image and text context, without any additional Tips. | `Above is the context {} of the target {}.`
`Please answer the question {} based on the`
`context and image. The output format must`
`strictly follow:`
`Answer: xxx` |
| Text-Image QA with One Tip | Generates a response to a question based on an image and text context, with a single Tip. | `The above is the context {} of the target {}.`
`This is a Tip: {} (which may not be correct).`
`Please answer the question {} based on the`
`context and image. The output format must`
`strictly follow:`
`Answer: xxx` |
| Text-Image QA with Multiple Tips | Generates a response to a question based on an image and text context, with multiple ranked Tips. | `The above is the context {} of the target`
`{}. These are the Tips (which may not be`
`correct):`
`Please answer the question {} based on the`
`context and image. The output format must`
`strictly follow:`
`Answer: xxx` |
| Text-Image QA with Bounding Boxes (No Tips) | Generates a response to a question based on an image, text context, and bounding box overlays, without any additional Tips. | `Above is the context {} of the target {}`
`document,`
`Please answer the question {},`
`Based on the context and image,`
`The output format strictly follows:`
`Answer: xxx` |
| Text-Image QA with Bounding Boxes (One Tip) | Generates a response to a question based on an image, text context, and bounding box overlays, with a single Tip. | `The above is the context {} of the target`
`{} document.`
`This is a Tip: {} (which may not be correct).`
`Please answer the question {},`
`Based on the context and image,`
`The output format strictly follows:`
`Answer: xxx` |
| Text-Image QA with Bounding Boxes (Multiple Tips) | Generates a response to a question based on an image, text context, and bounding box overlays, with multiple ranked Tips. | `The above is the context {} of the target`
`{} document.`
`These are Tips: {}, (which may not be`
`correct.)`
`Please answer the question {},`
`Based on the context and images,`
`The output format strictly follows:`
`Answer: xxx` |

# D    COMPUTATIONAL COST

## D.1    WARMER TUNING COST

Table S5 presents the training and inference resource consumption across five benchmark datasets with a consistent batch size of 16. The GPU memory usage remains within a reasonable range (approximately 25.5GB–28GB), demonstrating the framework's efficiency and scalability on standard hardware. The structural and semantic training times per epoch are well-balanced, typically ranging from 2 to 8 minutes, depending on dataset complexity. Notably, the inference time remains minimal—under 2.5 minutes for all datasets—highlighting the framework's practical deployment potential. These results indicate that the proposed framework achieves a favorable trade-off between training cost and performance, making it suitable for both research and real-world applications.

Table S5: Per-epoch GPU consumption and time cost across different datasets with a fixed batch size of 16. The times correspond to the most effective training configurations: 2 epochs for structural adaptation and 10 epochs for semantic adaptation.

| Dataset | Batch Size | GPU Consumption | Structural Time (1 Epoch) | Semantic Time (1 Epoch) | Inference Time |
|---|---|---|---|---|---|
| FormNLU-P | 16 | 27983.4M | 00:03:46 | 00:03:08 | 00:01:10 |
| FormNLU-H | 16 | 25736.0M | 00:03:58 | 00:03:01 | 00:01:02 |
| CORD | 16 | 26174.5M | 00:04:30 | 00:04:02 | 00:02:01 |
| EPHOIE | 16 | 27993.1M | 00:06:01 | 00:03:12 | 00:01:14 |
| FUNSD | 16 | 25566.2M | 00:08:10 | 00:02:01 | 00:00:59 |

## D.2    SYNDOC TIME LATENCY

The results in Table S6 demonstrate that SynDoc's recursive inference mechanism introduces only a modest increase in runtime while delivering consistent accuracy gains across datasets. Although the best-performing configuration requires slightly longer inference times—ranging from approximately 1.5× to 2× the latency of the vanilla setup—the improvement in ANLS is both stable and meaningful, particularly on FormNLU-P (+4.2 ANLS) and CORD (+0.8 ANLS). Importantly, most samples complete after a single iteration, keeping the overhead manageable. This lightweight inference cost stands in sharp contrast to traditional document KIE pipelines that rely on extensive manual annotation or expensive LLM/MLLM fine-tuning. By shifting complexity from heavy training to efficient inference, SynDoc provides a resource-efficient strategy that avoids knowledge leakage and tuning conflicts while achieving stronger zero-shot extraction performance.

Table S6: Comparison of runtime (mm:ss) and ANLS across configurations.

| Config | FormNLU-P | | FormNLU-H | | CORD | | EPHOIE | |
|---|---|---|---|---|---|---|---|---|
| | Time | ANLS | Time | ANLS | Time | ANLS | Time | ANLS |
| Vanilla | 15:24 | 76.09 | 17:22 | 66.86 | 06:26 | 84.35 | 35:25 | 81.82 |
| Best Performed | 30:47 | 80.29 | 31:17 | 67.73 | 11:27 | 85.19 | 84:17 | 82.15 |

# E    ADDITIONAL EVALUATION RESULTS

## E.1    VARIOUS PROMPT MODALITY PERFORMANCE

We present results from various prompting methods applied to baseline MLLMs and the Gemini-based SynDoc framework (Table S7). The findings indicate that multimodal prompting, which integrates OCR-extracted textual context with document images, generally enhances performance. However, the OCR Challenging dataset exhibits difficulties in some instances. For image-only prompting, some open-source models perform relatively poorly. Consequently, our SynDoc framework adopts the Image + Text context prompt as the primary approach for overall evaluation and ablation studies.

Table S7: Performance comparison of various models on different datasets.

| Models | Prompt | Formnlu-P | Formnlu-H | CORD | Ephoie | Funsd |
|---|---|---|---|---|---|---|
| InternVL2 | Context-only | 59.65 | 7.16 | 44.00 | 54.39 | 53.48 |
| Qwen2-VL | | 72.12 | 10.04 | 65.20 | 61.59 | 68.87 |
| Idefics2 | | 28.52 | 3.33 | 4.33 | 8.90 | 21.98 |
| GPT-4o | | 71.64 | 1.45 | 69.88 | 59.78 | 68.71 |
| Gemini | | 70.88 | 5.91 | 71.53 | 59.94 | 68.21 |
| InternVL2 | Image-only | 68.28 | 48.85 | 62.86 | 63.92 | 74.85 |
| Qwen2-VL | | 79.17 | 55.35 | 75.85 | 83.79 | 83.06 |
| Idefics2 | | 46.97 | 35.64 | 51.54 | 2.97 | 58.48 |
| GPT-4o | | 74.81 | 56.51 | 77.63 | 62.23 | 80.32 |
| Gemini | | 79.78 | 66.29 | 81.48 | 76.07 | 83.79 |
| InternVL2 | Context + Image | 66.56 | 45.47 | 66.84 | 68.92 | 74.95 |
| Qwen2-VL | | 79.71 | 55.33 | 79.12 | 83.35 | 82.77 |
| Idefics2 | | 57.54 | 33.31 | 54.45 | 15.22 | 62.11 |
| GPT-4o | | 76.16 | 56.49 | 79.05 | 79.40 | 80.05 |
| Gemini | | 76.09 | 66.86 | 84.35 | 81.82 | 83.56 |
| SynDoc | Context + Image | 81.91 | 68.02 | 85.19 | 82.15 | 83.02 |
| SynDoc | Context + Image + bbox | 80.93 | 68.13 | 85.40 | 82.08 | 83.87 |

## E.2    VARIOUS PROMPTING AND RAG METHOD PERFORMANCE

To further validate the effectiveness of SynDoc, we conducted comprehensive comparisons against multiple prompting-based reasoning approaches (Chain-of-Thought, Self-Consistency, Reflexion) as well as sparse and dense RAG baselines. For prompting methods, Chain-of-Thought responses were generated using explicit reasoning instructions, while Self-Consistency and Reflexion were executed with a maximum of five iterative reasoning steps. All of these methods operate solely on the input context and the implicit perceptual and world knowledge of the MLLM, without any external adaptation. Across all datasets, SynDoc achieves consistently higher accuracy while requiring substantially less inference time compared to these iterative reasoning strategies, highlighting the efficiency of its recursive retrieval–generation interaction.

We additionally benchmarked traditional RAG pipelines using both sparse (TF-IDF, BM25) and dense retrievers-based on Sentence-BERT (Top-5 retrieval). These methods typically introduce noisy or irrelevant passages into the context window, which in turn degrades MLLM performance. This trend underscores a key advantage of SynDoc: the warmer, trained using synthetic structural and semantic adaptation signals, retrieves fine-grained and domain-grounded cues rather than raw passages. As a result, SynDoc mitigate the noise amplification commonly observed in classical RAG systems and provides stable, high-quality guidance to the MLLM throughout inference.

## E.3    MORE WARMER ABLATION STUDIES

The results in Table S9 show that the joint-grained framework generally outperforms both the fine-grained and coarse-grained variants across most datasets, demonstrating its stronger adaptation capability when semantic and structural cues are integrated. While the fine-grained span-based QA head performs poorly on all benchmarks due to its limited ability to generalize from synthetic annotations, the coarse-grained entity retriever provides a substantial improvement by leveraging more stable structural patterns in documents. The joint configuration further enhances performance on

Table S8: Comparison of ANLS and Runtime (mm:ss) Across Various Prompting and Retrieval Methods

| Model | FormNLU-P (F-P) | | FormNLU-H (F-H) | | CORD | | EPHOIE | | FUNSD | |
|---|---|---|---|---|---|---|---|---|---|---|
| | ANLS | Time | ANLS | Time | ANLS | Time | ANLS | Time | ANLS | Time |
| CoT | 77.78 | 15:11 | 65.67 | 17:48 | 84.56 | 06:23 | 81.17 | 35:25 | 83.12 | 10:24 |
| Self-Consistency | 75.78 | 44:59 | 67.56 | 72:51 | 83.99 | 33:11 | 81.52 | 174:11 | 82.98 | 60:11 |
| Reflexion | 78.43 | 45:21 | 68.19 | 75:13 | 85.11 | 40:32 | 81.23 | 191:45 | 84.01 | 56:25 |
| TF-IDF | 75.24 | 15:02 | 64.12 | 16:30 | 83.22 | 06:01 | 81.01 | 33:11 | 82.43 | 10:03 |
| BM25 | 75.45 | 16:34 | 64.72 | 17:21 | 83.48 | 06:32 | 80.82 | 37:32 | 83.00 | 10:21 |
| Dense | 74.25 | 15:43 | 65.95 | 17:44 | 83.51 | 06:22 | 81.74 | 34:24 | 81.77 | 09:22 |
| **SynDoc** | **80.29** | 30:47 | **67.73** | 31:17 | **85.19** | 11:27 | **82.15** | 84:17 | **83.87** | 21:23 |

Table S9: Impact of Grainularity on Domain Adaptation Performance (ANLS).

| Granularity | FormNLU-P | FormNLU-H | CORD | EPHOIE |
|---|---|---|---|---|
| Fine-grained Span-based QA Head | 38.72 | 15.50 | 47.21 | 20.52 |
| Coarse-grained Entity Retrieving Head | 65.35 | 25.55 | 63.17 | 29.74 |
| Joint (Fine + Coarse) | **66.19** | **28.29** | **66.25** | 27.16 |

FormNLU-P, FormNLU-H, and CORD, confirming that combining granularities enables richer and more complementary document understanding. The only exception is EPHOIE, where coarse-grained retrieval slightly surpasses the joint model—likely a result of limited Chinese-text modelling capacity in the LayoutLMv3-Chinese backbone, which restricts fine-grained semantic alignment. Overall, the trend validates that synthetic semantic/structural adaptation is most effective when both grain levels are jointly optimized.

### E.4 EFFECTIVENESS OF RECURSIVE INFERENCE.

Recursive inference is introduced to enhance both Warmer retrieval and the quality of MLLM-generated answers. Table S10 shows that models exhibit improved performance when more than one iteration is conducted. This demonstrates that Warmer and the LLM generator can mutually reinforce each other, enabling the model to generate more accurate final predictions. Additionally, we observed that open-source models (InternVL, QWenVL) typically require more iterations to reach peak performance, while the closed-source Gemini often achieves its best results with fewer iterations. Moreover, datasets that present OCR challenges (F-H and Ephoie) benefit from additional iterations, with all models requiring at least two iterations for optimal performance.

Table S10: Performance trends of iterative tuning. Int: InternVL2; QW: QWenVL2; Gemi: Gemini.

| Iter. | F-P | | | F-H | | | CORD | | | EPHOIE | | |
|---|---|---|---|---|---|---|---|---|---|---|---|---|
| | Int | Qw | Gemi | Int | QW | Gemi | Int | QW | Gemi | Int | QW | Gemi |
| Vani. | 66.56 | 78.05 | 76.09 | 45.47 | 43.65 | 66.86 | 66.84 | 77.86 | 84.35 | 68.92 | 70.36 | 81.82 |
| Iter 1 | 68.09 | 76.53 | **80.29** | 46.81 | 44.43 | 67.73 | **68.80** | 76.93 | **85.19** | 68.54 | 75.03 | 81.80 |
| Iter 2 | **70.12** | 77.22 | 80.17 | 46.17 | **45.27** | 67.60 | 67.89 | 76.70 | 84.67 | 69.49 | **75.55** | **81.91** |
| Iter 3 | 68.54 | 76.75 | 80.15 | **47.23** | 44.50 | 67.32 | 67.29 | 76.93 | 84.65 | **70.24** | 75.44 | 81.71 |
| Iter 4 | 68.28 | **77.27** | 79.88 | 45.54 | 45.26 | **67.63** | 66.84 | 76.70 | 84.39 | 68.99 | 75.55 | 82.15 |
| Iter 5 | 70.21 | 76.75 | 80.06 | 44.86 | 44.51 | 67.63 | 67.28 | **76.93** | 84.40 | 70.07 | 75.44 | 81.86 |

### E.5 MORE DETAILED EXPERIMENTAL RESULTS

We provide detailed experimental results for different configurations of MLLM inference, from Table S11 to Table S13.

Table S11: Performance comparison across iterations for different models on the CORD, Printed, Handwritten, Ephoie, and FUNSD dataset with Top-1 warmer retrieved entity.

| Model | Baseline | | Iteration 1 | | Iteration 2 | | Iteration 3 | | Iteration 4 | | Iteration 5 | |
|---|---|---|---|---|---|---|---|---|---|---|---|---|
| | Warmer | LLM | Warmer | LLM | Warmer | LLM | Warmer | LLM | Warmer | LLM | Warmer | LLM |
| **CORD** | | | | | | | | | | | | |
| InternVL | 67.26 | 66.84 | 63.38 | 68.80 | 57.74 | 67.89 | 58.50 | 67.29 | 59.70 | 66.84 | 57.82 | 67.28 |
| QWenVL (2B) | 67.26 | 12.17 | 63.38 | 16.36 | 59.75 | 16.75 | 59.86 | 16.43 | 59.75 | 16.75 | 59.86 | 16.43 |
| QWenVL (7B) | 67.26 | 77.86 | 63.38 | 76.93 | 59.89 | 76.70 | 59.64 | 76.93 | 59.89 | 76.70 | 59.64 | 76.93 |
| QWenVL (72B) | 67.26 | 79.12 | 63.38 | 78.02 | 59.98 | 77.96 | 60.30 | 77.81 | 59.98 | 77.96 | 60.30 | 77.81 |
| Gemini | 67.26 | 84.35 | 63.37 | 85.19 | 64.15 | 84.67 | 64.32 | 84.65 | 64.32 | 84.39 | 64.04 | 84.40 |
| **Printed** | | | | | | | | | | | | |
| InternVL | 66.19 | 66.56 | 73.57 | 68.09 | 68.65 | 70.12 | 70.32 | 68.54 | 69.20 | 68.28 | 69.19 | 70.21 |
| QWenVL (2B) | 66.19 | 44.85 | 73.57 | 50.34 | 61.63 | 50.45 | 61.82 | 50.52 | 61.76 | 50.48 | 61.84 | 50.54 |
| QWenVL (7B) | 66.19 | 78.05 | 73.57 | 76.53 | 72.52 | 77.22 | 73.18 | 76.75 | 72.61 | 77.27 | 73.18 | 76.75 |
| QWenVL (72B) | 66.19 | 79.71 | 73.57 | 81.21 | 74.41 | 81.42 | 74.54 | 81.20 | 74.58 | 81.42 | 74.54 | 81.20 |
| Gemini | 66.19 | 76.09 | 73.57 | 80.29 | 73.76 | 80.17 | 73.72 | 80.15 | 73.76 | 79.88 | 73.60 | 80.06 |
| **Handwritten** | | | | | | | | | | | | |
| InternVL | 31.64 | 45.47 | 38.11 | 46.81 | 32.29 | 46.17 | 32.70 | 47.23 | 32.06 | 45.54 | 32.76 | 44.86 |
| QWenVL (2B) | 31.64 | 14.56 | 38.11 | 19.21 | 24.95 | 19.33 | 25.45 | 19.19 | 25.02 | 19.36 | 25.44 | 19.20 |
| QWenVL (7B) | 31.64 | 43.65 | 38.11 | 44.43 | 34.51 | 45.27 | 35.25 | 44.50 | 34.83 | 45.26 | 35.26 | 44.51 |
| QWenVL (72B) | 31.64 | 55.33 | 38.11 | 58.33 | 38.37 | 58.40 | 38.33 | 58.37 | 38.48 | 58.58 | 38.36 | 58.37 |
| Gemini | 31.64 | 66.86 | 38.11 | 67.73 | 38.79 | 67.60 | 39.15 | 67.32 | 38.84 | 67.63 | 38.92 | 67.63 |
| **Ephoie** | | | | | | | | | | | | |
| InternVL | 27.16 | 68.92 | 27.98 | 68.54 | 25.78 | 69.49 | 25.94 | 70.24 | 26.00 | 68.99 | 25.96 | 70.07 |
| QWenVL (2B) | 27.16 | 46.13 | 27.98 | 36.51 | 27.10 | 37.00 | 26.78 | 36.39 | 27.10 | 36.97 | 26.78 | 36.39 |
| QWenVL (7B) | 27.16 | 70.36 | 27.98 | 75.03 | 26.79 | 75.55 | 26.76 | 75.44 | 26.79 | 75.55 | 26.76 | 75.44 |
| QWenVL (72B) | 27.16 | 83.35 | 27.98 | 81.95 | 26.38 | 82.08 | 26.51 | 82.06 | 26.38 | 82.08 | 26.51 | 82.06 |
| Gemini | 27.16 | 81.82 | 27.98 | 81.80 | 25.94 | 81.91 | 26.03 | 81.71 | 25.94 | 82.15 | 26.12 | 81.86 |
| **FUNSD** | | | | | | | | | | | | |
| InternVL | 61.24 | 74.95 | 59.64 | 73.18 | 58.30 | 72.13 | 58.44 | 73.41 | 58.97 | 73.57 | 58.98 | 73.12 |
| QWenVL | 61.24 | 79.12 | 61.94 | 74.84 | 60.03 | 75.73 | 60.92 | 74.57 | 59.93 | 75.73 | 60.92 | 74.57 |
| Gemini | 61.24 | 83.56 | 59.17 | 82.77 | 59.77 | 83.02 | 60.06 | 82.38 | 59.54 | 82.91 | 60.11 | 82.36 |

Table S12: Top-3 Warmer Retrieved Entity Performance comparison across iterations for different models on the CORD, Printed, Handwritten, and Ephoie dataset.

| Model | Baseline | | Iteration 1 | | Iteration 2 | | Iteration 3 | | Iteration 4 | | Iteration 5 | |
|---|---|---|---|---|---|---|---|---|---|---|---|---|
| | Warmer | LLM | Warmer | LLM | Warmer | LLM | Warmer | LLM | Warmer | LLM | Warmer | LLM |
| **CORD** | | | | | | | | | | | | |
| InternVL | 67.26 | 66.84 | 63.38 | 61.61 | 60.19 | 65.31 | 53.73 | 64.75 | 53.52 | 61.70 | 54.06 | 62.22 |
| QWenVL | 67.26 | 77.86 | 63.38 | 78.16 | 59.65 | 77.96 | 59.34 | 78.12 | 59.65 | 77.96 | 59.34 | 78.12 |
| Gemini | 67.26 | 84.35 | 63.38 | 83.46 | 63.79 | 82.34 | 63.42 | 83.07 | 63.42 | 83.07 | 63.69 | 83.00 |
| **Printed** | | | | | | | | | | | | |
| InternVL | 66.19 | 66.56 | 73.56 | 65.91 | 67.70 | 67.85 | 67.55 | 67.12 | 68.53 | 66.21 | 66.92 | 66.38 |
| QWenVL | 66.19 | 78.05 | 73.57 | 77.08 | 72.93 | 76.60 | 72.81 | 76.63 | 72.53 | 76.72 | 72.80 | 76.67 |
| Gemini | 66.19 | 76.09 | 73.99 | 81.60 | 74.12 | 81.91 | 74.30 | 81.63 | 74.01 | 81.58 | 74.28 | 81.46 |
| **Handwritten** | | | | | | | | | | | | |
| InternVL | 31.64 | 45.47 | 38.11 | 43.48 | 32.64 | 43.24 | 30.92 | 42.02 | 31.75 | 43.15 | 31.93 | 43.52 |
| QWenVL | 31.64 | 43.65 | 38.11 | 42.03 | 33.65 | 43.37 | 33.68 | 41.66 | 32.60 | 42.62 | 33.28 | 41.55 |
| Gemini | 31.64 | 66.86 | 38.11 | 66.82 | 39.35 | 67.68 | 39.48 | 67.12 | 39.15 | 66.80 | 38.79 | 67.49 |
| **Ephoie** | | | | | | | | | | | | |
| InternVL | 27.16 | 68.92 | 27.98 | 70.29 | 26.00 | 69.04 | 26.08 | 69.35 | 26.17 | 68.15 | 26.30 | 69.41 |
| QWenVL | 27.16 | 70.36 | 27.98 | 73.91 | 26.36 | 74.29 | 26.68 | 74.18 | 26.28 | 74.29 | 26.68 | 74.18 |
| Gemini | 27.16 | 81.82 | 27.98 | 81.18 | 26.23 | 81.13 | 26.25 | 81.16 | 26.27 | 81.43 | 26.10 | 81.32 |

Table S13: Top-5 Performance comparison across iterations for different models on the CORD, Printed, Handwritten, and Ephoie dataset.

| Model | Baseline | | Iteration 1 | | Iteration 2 | | Iteration 3 | | Iteration 4 | | Iteration 5 | |
|---|---|---|---|---|---|---|---|---|---|---|---|---|
| | Warmer | LLM | Warmer | LLM | Warmer | LLM | Warmer | LLM | Warmer | LLM | Warmer | LLM |
| **CORD** | | | | | | | | | | | | |
| InternVL | 67.26 | 66.84 | 63.37 | 64.25 | 57.02 | 63.21 | 54.63 | 68.18 | 55.93 | 66.76 | 57.87 | 65.13 |
| QWenVL | 67.26 | 77.86 | 63.38 | 78.20 | 59.54 | 77.49 | 58.91 | 78.44 | 60.08 | 77.53 | 58.91 | 78.16 |
| Gemini | 67.26 | 84.35 | 63.38 | 84.57 | 63.79 | 82.85 | 63.99 | 83.37 | 63.79 | 82.77 | 63.79 | 83.39 |
| **Printed** | | | | | | | | | | | | |
| InternVL | 66.19 | 66.56 | 73.57 | 66.88 | 69.19 | 66.17 | 67.46 | 65.23 | 65.67 | 65.67 | 66.27 | 66.27 |
| QWenVL | 66.19 | 78.05 | 73.57 | 76.35 | 72.22 | 77.01 | 72.66 | 76.67 | 72.34 | 77.27 | 72.70 | 76.21 |
| Gemini | 66.19 | 76.09 | 73.58 | 80.10 | 73.35 | 80.35 | 73.70 | 80.20 | 73.40 | 80.36 | 73.54 | 80.08 |
| **Handwritten** | | | | | | | | | | | | |
| InternVL | 31.64 | 45.47 | 38.11 | 43.82 | 32.79 | 44.13 | 33.66 | 43.78 | 31.87 | 41.55 | 32.11 | 43.22 |
| QWenVL | 31.64 | 43.65 | 38.11 | 40.12 | 32.51 | 41.97 | 33.13 | 40.18 | 32.26 | 41.75 | 32.99 | 39.78 |
| Gemini | 31.64 | 66.86 | 38.11 | 66.90 | 39.05 | 67.33 | 39.06 | 67.51 | 38.99 | 67.01 | 39.11 | 68.02 |
| **Ephoie** | | | | | | | | | | | | |
| InternVL | 27.16 | 68.92 | 27.98 | 68.88 | 26.18 | 68.66 | 25.93 | 67.90 | 26.06 | 69.61 | 25.87 | 69.77 |
| QWenVL | 27.16 | 70.36 | 27.98 | 74.32 | 26.32 | 74.32 | 26.56 | 74.35 | 26.32 | 74.34 | 26.67 | 74.24 |
| Gemini | 27.16 | 81.82 | 27.98 | 81.33 | 26.35 | 81.18 | 26.31 | 81.58 | 26.37 | 81.23 | 26.28 | 81.45 |

# F ADDITIONALY CASE STUDIES

Figures S1 and S2 present qualitative case studies from the CORD and Ephoie datasets, respectively, highlighting the complementary strengths of the MLLM-based self-correction pipeline and the Top-K retrieval. In Figure S1, the MLLM initially predicts "24,000", and the Warmer module retrieves a noisy string "Qty=4.00240.000". Despite the noise, the final MLLM module correctly interprets the answer as "4", demonstrating its robustness to OCR errors and its ability to reason over imperfectly retrieved content. In Figure S2, a query about a student's name is given, where the initial MLLM output is incorrect. However, the Warmer module retrieves relevant entities and ranks the correct answer within the Top-3, enabling the final MLLM stage to recover the accurate result. These examples collectively demonstrate the pipeline's effectiveness in overcoming early-stage retrieval errors and OCR-related noise in complex document QA tasks.

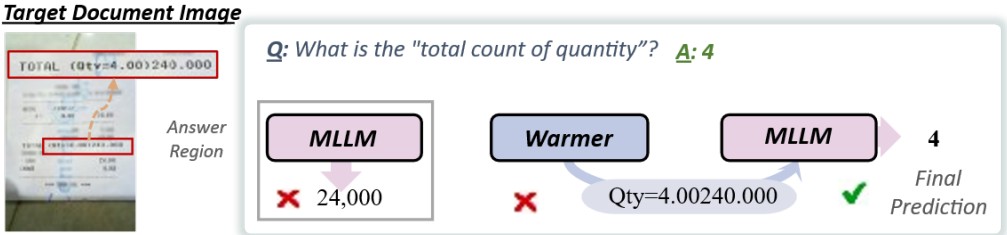

Figure S1: Qualitative case studies about the CORD dataset for demonstrating the effectiveness of Warmer in retrieving the content and the MLLM self-correction ability for OCR errors.

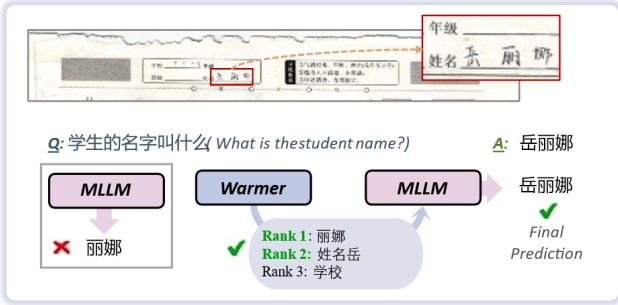

Figure S2: Qualitative case studies about the Ephoie dataset for demonstrating the effectiveness of Top-$K$.

# G LLM USAGE

Large Language Models (LLMs) were used to aid in the writing and polishing of the manuscript. Specifically, we used an LLM to refine the language, improve readability, and ensure clarity across various sections of the paper. The model helped with tasks such as sentence rephrasing, grammar checking, and enhancing the overall flow of the text.

It is important to note that the LLM was not involved in the ideation, research methodology, or experimental design. All research concepts, ideas, and analyses were developed and conducted by the authors. The contributions of the LLM focused solely on improving the paper's linguistic quality, with no involvement in the scientific content or data analysis.

The authors take full responsibility for the content of the manuscript, including any text generated or polished by the LLM. We have ensured that the LLM-generated text adheres to ethical guidelines and does not contribute to plagiarism or scientific misconduct.

