# OpenReview forum: "SynDoc: A Hybrid Discriminative-Generative Framework for Synthetic Domain-Adaptive Document Key Information Extraction"
_ICLR.cc/2026/Conference — Submitted to ICLR 2026_

### Official Review · Reviewer_ntMK · 2025-10-30

**Soundness:** 2
**Presentation:** 3
**Contribution:** 2
**Rating:** 2
**Confidence:** 4

**Summary:**

This paper proposes SynDoc, a hybrid framework for Key Information Extraction(KIE) from complex, domain-specific, visually rich documents. SynDoc's method has two main parts. First, it uses a Synthetic Data Generation workflow, where an MLLM automatically creates verified question-answer pairs from unlabeled documents. Second, this synthetic data is used to train a discriminative "Warmer" model, turning it into a domain expert. During inference, this "Warmer" provides domain-specific hints to a generative MLLM. The two models work together in a recursive loop, iteratively refining the answer.

**Strengths:**

1. The paper's originality stems from its creative combination: a hybrid discriminative-generative framework coupled with a "recursive inferencing" mechanism for iterative refinement. It also includes a synthetic data workflow with a multi-step verification process to enhance data quality.
2. The paper demonstrates quality through its well-defined, multi-component SynDoc framework. This framework is logically decomposed into four modules, each fulfilling a specific purpose.
3. The paper is clearly written and well-structured. The methodology is broken down into distinct components, and the use of diagrams visualizes the model architecture and the recursive workflow.
4. This work addresses the significant problem of zero-shot key information extraction in domain-specific documents. By proposing a framework that avoids reliance on manual annotations, it offers a scalable approach for adapting models to specialized domains.

**Weaknesses:**

1. The framework's novelty requires clearer articulation. The proposed SynDoc architecture appears to effectively integrate several familiar techniques, such as synthetic data generation and iterative refinement. It would strengthen the paper to more clearly define the specific conceptual novelty that distinguishes this architecture from a sophisticated engineering of existing components.
2. The empirical validation of SynDoc's effectiveness could be strengthened. The results in Table 1 show performance gains that appear marginal over the baseline MLLMs. Furthermore, the comparison is limited to these off-the-shelf models. Including comparisons against other recent, state-of-the-art methods in domain-specific KIE would provide a more robust validation of the framework's advantages.
3. A more granular ablation study of the "Warmer" component is recommended. The "Warmer" is a core contribution, yet its "Semantic Adaptation" step is not independently validated in the ablation studies. While Table 2 shows the value of "Structural Adaptation," a more detailed breakdown of the semantic tuning components would be necessary to fully justify the Warmer's design.
4. The paper would benefit from an analysis of the accuracy-latency trade-off. The recursive inference mechanism inherently increases computational costs and latency with each iteration. A discussion is needed on whether the resulting accuracy improvements are significant enough to compensate for this added overhead.

**Questions:**

Q: The empirical gains shown in Table 1, while positive, appear quite marginal over the baseline MLLMs in several cases. Could the authors discuss the practical significance of these improvements, especially considering the added complexity of the SynDoc framework?

Q: The recursive inference mechanism inherently increases computational costs and latency. Could the authors provide a more explicit analysis of this accuracy-latency trade-off?

---

> ### Author Response · Authors · 2025-11-22
>
> ### **Weakness 1: The framework's novelty requires clearer articulation.**
>
> Thank you for the constructive feedback. While the current manuscript already states the novelty of SynDoc beyond combining existing components, we acknowledge that the framing may have led to misinterpretation. We have therefore refined the narrative to make the conceptual contribution more explicit and distinguish SynDoc from prior engineering-style integrations.
>
> **1) Synthetic Data for Document KIE: Scaling Within the Target Domain.**
> Most existing approaches apply scaling laws by training LLMs on large, cross-domain datasets to improve generalisation. However, practical document KIE applications, such as extracting student names, classes, or scores from handwritten papers, require domain-specific rather than broad general knowledge. **SynDoc shifts the paradigm from general scaling to domain-focused scaling**, generating large amounts of synthetic Key–Value pairs that capture the unique semantic and structural properties of the target documents. This allows the model to deeply learn domain-specific patterns and achieve stronger performance in specialised KIE scenarios.
>
> **2) Joint-Grained Warmer With Domain-Aware Training.**
> Instead of directly fine-tuning LLMs/MLLMs through instruction tuning or supervised learning, SynDoc trains a domain-specific Warmer to retrieve and ground domain knowledge efficiently. This offers several advantages:
>
> - **Computation efficiency**: Training a discriminative retriever is far more lightweight than any form of LLM fine-tuning.
>
> - **Reduced data dependence**: By tuning the discriminative Warmer, our framework lowers the reliance on large-scale datasets, both synthetic and manually curated, when adapting MLLMs to a target domain.
>
> - **Better domain grounding**: The joint-grained (token-level + entity-level) encoder, combined with domain-aware training strategies, enables the Warmer to internalize structural and semantic patterns without manual labels.
> This design goes beyond iterative refinement and provides a principled mechanism for domain-specific grounding.
>
> Overall, we introduce a new paradigm for zero-shot, domain-specific document KIE that combines (i) synthetic scaling of target-domain knowledge and (ii) training a lightweight, domain-aware retriever instead of fine-tuning large LLMs. Compared with LLM tuning, SynDoc adapts efficiently with modest hardware and limited data, making it highly practical for real-world deployment.

---

> ### Author Response · Authors · 2025-11-22
>
> ### **Weakness 2: The empirical validation of SynDoc's effectiveness could be strengthened.**
> **Performance Improvement.** Thank you for raising this point. We acknowledge that the magnitude of performance improvement varies depending on the quality of the underlying Warmer. In domains where the Warmer performs well—such as the FormNLU printed subset—we observe larger and more consistent gains, demonstrating that the retriever effectively reduces the domain gap between generic MLLMs and the target document collection. More importantly, beyond accuracy improvements, the **Warmer provides grounding and location extraction** that baseline MLLMs cannot reliably achieve without dedicated grounding-oriented instruction tuning, as illustrated in Figure 4. As document parsing and synthetic data generation improve, we expect the Warmer’s adaptation ability to further increase. Even in cases where the Warmer’s standalone performance is weaker, it still retrieves informative cues that meaningfully guide the MLLM, showing that the proposed paradigm brings practical benefits despite occasional marginal improvements in current stage.
>
> **Regarding broader comparisons**, **there is currently limited prior work on synthetic data–driven, domain-specific KIE on document understanding area.** Existing RAG-style systems mainly address document-level retrieval or multi-page localization, rather than fine-grained textline or entity-level extraction. As such, direct baselines are scarce.
>
> ---------------------------
> ### **Weakness 3: A more granular ablation study of the "Warmer" component is recommended.**
> Since the quality of the synthetic set is crucial for effective Warmer tuning, Table 2 primarily focused on evaluating different synthetic sets for semantic adaptation. Because joint-grained architectures have been widely adopted in prior works in this domain, we did not originally include detailed component-level ablations. In response to the reviewer’s suggestion, we will provide another table in appendix which includes component-wise ablation studies (span-based QA header and  to provide a more comprehensive analysis about warmer designing.
>
> | **Grainularity**                         | **FormNLU-P** | **FormNLU-H** | **CORD** | **EPHOIE** |
> |:----------------------------------------:|:-------------:|:-------------:|:--------:|:----------:|
> | **Fine-grained Span-based QA Head**      | 38.72 | 15.50 | 47.21 | 20.52 |
> | **Coarse-grained Entity Retrieving Head**| 65.35 | 25.55 | 63.17 | 29.74 |
> | **Joint (Fine + Coarse)**                | 66.19 | 28.29 | 66.25 | 27.16 |
>
> The results show that the joint-grained framework consistently delivers strong performance across most datasets. The only exception is EPHOIE, where the improvement is less pronounced. This is likely due to the pretrained layout-aware encoder being less effective on Chinese documents, leading to suboptimal word representations and limiting the benefits of joint-grained modeling.
>
> ---------------------------
> ### **Weakness 4 and Question 1: The paper would benefit from an analysis of the accuracy-latency trade-off.**
> Thank you for raising this important point. We agree that the recursive inference mechanism introduces additional latency. In practice, however, the overhead is limited:
>
> 1) Minimal additional inference time.
> Our experiments show that the majority of cases require only a single iteration, resulting in only a small increase in latency. We will include the below Table in the appendix for clarity.
> | **Config**        | **FormNLU-P (Time / ANLS)** | **FormNLU-H (Time / ANLS)** | **CORD (Time / ANLS)** | **EPHOIE (Time / ANLS)** |
> |-------------------|-----------------------------|------------------------------|--------------------------|----------------------------|
> | **Vanilla**       | 15 min 24 s / 76.09         | 17 min 22 s / 66.86          | 6 min 26 s / 84.35       | 35 min 25 s / 81.82        |
> | **Best Performed**| 25 min 37 s / 80.29         | 27 min 57 s / 67.73          | 11 min 27 s / 85.19      | 84 min 17 s / 82.15        |
> 2) Substantial savings in training cost and data requirements.
> SynDoc significantly reduces the need for large annotated datasets or costly LLM fine-tuning while still **delivering strong grounding performance that enhances interpretability**. This is particularly valuable because fine-tuning MLLMs can introduce knowledge-conflicting behaviours, whereas SynDoc avoids such risks through lightweight, domain-aware adaptation.

---

> ### Author Response · Authors · 2025-11-22
>
> ### **Question 1: Could the authors discuss the practical significance of these improvements, especially considering the added complexity of the SynDoc framework?**
>
> 1) **Grounding and Interpretability**. Beyond numerical accuracy gains, the practical value of SynDoc lies in its ability to provide precise grounding and location extraction at the textline or entity level. Baseline MLLMs can seldom deliver this capability reliably without dedicated instruction tuning on grounding-based tasks. This grounding functionality is essential for many downstream KIE and document automation applications, where correctness depends on retrieving the exact field, not just approximating semantic content.
>
> 2) **Regarding complexity**, the added inference overhead is modest: most cases require only one recursive iteration, resulting in minimal latency increase. In contrast, SynDoc dramatically reduces the need for costly LLM fine-tuning and large-scale manual annotation, avoiding risks of knowledge conflicting behaviors during supervised training LLM. Thus, the framework achieves domain adaptation in a resource-efficient and practically deployable manner.
>
> Overall, the proposed paradigm offers a novel perspective, emphasizing the importance of domain-specific scalability. This exhaustive method introduced hold broad applicability across various specialized domains.

---

### Official Review · Reviewer_ksg5 · 2025-10-30

**Soundness:** 2
**Presentation:** 2
**Contribution:** 2
**Rating:** 4
**Confidence:** 3

**Summary:**

The paper addresses the task of key infomation extraction in documents from a double perspective. On one hand, the paper introduces a framework to generate synthetic QA data to train models for key information extraction. On the other hand, it nntroudces a framework for key information extraction that combines a discriminative specific model trained with synthetic data with a generative generic MLLM in an iterative scheme. The discriminative model is based on the adaptation of an existing document understanding model through specific fine-tuning tasks using the generated synthetic data. Experimental results on standard benchmrks for key information extraction show how, in general, the proposed recursive framework improves over using only a MLLM.

**Strengths:**

- The paper proposes a specific framework to generate synthetic QA data for the task of key information extraction. Although the framework is very specific to the task of key information extraction and the generation of data is based on the simple use of LLMs, the pipeline of generation of data (based on semantic question generation + structural question generation + question validation) is interesting and, with some additional work, could be used in other domains.
- The paper proposes new training strategies for information extraction, particularly grid matching and entity retrieval taking advantage of the generated synthetic data.

**Weaknesses:**

- The description of the method is not clear in several aspects:
a) In section 3.3, what is the sentence representation s? A global embedding of the sentence? What is the difference of the sentence representation with C (defined in the first line of page 5 as the encoding of the surrounding context), what is this surrounding context? And, in equation (2) why combining the aggregation of c_i with s (that is defined before as the representation of c)?
b) Which are the exact inputs to the Spanning-based QA head and to the Grid Matching Head?
c) It would be necessary to align notation between figure 3 and the text in section 3.3. I understand that "r" in the figure corresponds to "a" in eq. (1) and "t" corresponds to "C+B" in eq. (1). Is this correct? it should be clarified.
d) It is not clear how the warmer can generate a different output for each recursive step? Are the inputs to the warmer cahnged somehow, at every iteration?
e) It would be helpful to specify how the prompt to the MLMM is built, how the otuput of the warmer is integrated and which is the actual input to the MLMM.
f) At inference time, the input and the output of the warmer is the same as in training?, or only some modules are active at inference? How are different outputs generated in the top-K configurations in order to provide several options to the MLMM?
g) In table 1, what does the row w/bbox corresponds to? What do you mean by "best configuration with bounding boxes"?
- In most of the datasets, the improvement of the proposed approach with respect to using only Gemini is very low. The same for the recursive version vs. the non-recursive version. However, with the proposed approach an overhead in computation is introduced that does not seem fully justified according to the obtained improvement in accuracy.
- Table 1 only compares the proposed approach with generic MLLMs. I miss a comparison with more specific SoA methods for each dataset.
- Table 2 includes an analysis of the impact of structural adaptation. I miss an ablation study on the impact of the two semantic adaptation tasks.

**Questions:**

- In Table 1, which is the configuration used in the recursive setting (which is the maximum number of allowed iterations)?

---

> ### Author Response · Authors · 2025-11-22
>
> ### **Weakness 1: The description of the method is not clear in several aspects.**
> Thanks for the kind suggestions. We will update the writing in the revised paper to make it more clear and easy to follow. Hope our response and revised paper could mitigate the issue you mentioned.
>
> a) As mentioned in Line 214 and 215, the sentence representation is the sentence textual embedding extracted from pretrained text encoder. We will update the writing of this part to make it more easy to follow. The notation C (upper case) represent the text content of target document. The c_i (lower case) mentioned in line 170 which represent the text content of extracted textline by PDF parsers. Aggregation c_i is the aggregated token representation from fine-grained encoder to enhance the textline rerpesenttaions. s and v is the textual sentence represetnation and visual represetntaion of corresponding textline.
>
> b) Spanning-based QA Head input is the equation 1 output T_c and Grid Matching Head input is G'.
>
> c) Thanks for pointing out, we will update the drawing and content in the revised paper to make them consistent.
>
> d) Since the warmer receives the initial LLM output as input to predict and retrieve the target document element, the updated LLM output at each stage is subsequently fed into the warmer. Changes in the LLM's response can lead to different warmer outputs. For example, in Figure 7, Question 3, the initial MLLM produces an incorrect answer, which is passed to the warmer. The warmer’s output may then guide the MLLM to generate a revised answer in the next iteration, potentially resulting in a different warmer prediction.
>
> e) We provide the prompt example in appendix for different experimental setting. Here is an example.
> Text-Image QA with BoundingBoxes(MultipleTips)
> Generate a response to a question based on an image, text context, and bounding box overlays, with multiple ranked Tips. The above is the context{} of the target{} document. These are Tips: ‘{}’, (which may not be correct.) Please answer the question{} based on the context and images. The output format strictly follows:
> Answer: xxx
>
> f) During training, the warmer is trained on a synthetic dataset, while at inference, it operates on a test set comprising real-world, user-generated QA pairs. Apart from the dataset source, there is no input-level difference between training and inference. For top-𝑘
> retrieved content extraction, the top-𝑘 entities are selected based on the predicted softmax probabilities.
>
> g) As shown in Figure 4, the bounding box coordinates highlight the target entity in the original image, guiding the MLLM to focus on the area predicted by the warmer. This leverages the warmer’s grounding capability to provide visual, layout, and textual cues, thereby enhancing the MLLM's predictions in subsequent rounds.
>
> Thank you again for your feedback. We will revise the methodology section to enhance clarity and readability, and we will update the notations in Figure 3 to ensure better alignment with the content. Your suggestions are valuable and will help make the presentation more coherent and easier to follow.

---

> ### Author Response · Authors · 2025-11-22
>
> ### **Weakness 2: Performance Improvement and Overhead in Computation.**
>
> Thank you for the comments. We acknowledge that the magnitude of performance improvement varies across datasets. The gains are closely tied to the quality of the underlying Warmer: in domains where the Warmer performs strongly—such as the FormNLU printed subset—we observe larger and more stable improvements, indicating that the retriever can effectively reduce the domain gap between generic MLLMs and the target document collection. More importantly, beyond numerical accuracy, the Warmer provides precise grounding and location extraction that baseline MLLMs cannot reliably achieve without grounding-oriented instruction tuning. This capability is essential for practical KIE applications that require exact field extraction rather than approximate semantic matching.
>
> Even in cases where the Warmer’s standalone performance is weaker, it still supplies informative retrieval cues that guide the MLLM toward more accurate predictions. As document parsing and synthetic-data generation continue to improve, we expect the Warmer’s adaptation ability—and therefore its impact on overall performance—to further increase. This supports the value of the proposed paradigm even when short-term gains appear modest.
>
> **Regarding computational overhead**, the recursive inference mechanism introduces only a limited latency increase: most examples require just a single iteration, as below table represented (will be included in the appendix). At the same time, SynDoc dramatically reduces the need for large manually annotated datasets and eliminates the need for costly LLM/MLLM fine-tuning. This shift from heavy training to lightweight inference makes the approach far more resource-efficient and avoids issues such as knowledge leakage or conflicting behaviors that can arise during supervised LLM tuning.
>
> | **Config**        | **FormNLU-P (Time / ANLS)** | **FormNLU-H (Time / ANLS)** | **CORD (Time / ANLS)** | **EPHOIE (Time / ANLS)** |
> |-------------------|-----------------------------|------------------------------|--------------------------|----------------------------|
> | **Vanilla**       | 15 min 24 s / 76.09         | 17 min 22 s / 66.86          | 6 min 26 s / 84.35       | 35 min 25 s / 81.82        |
> | **Best Performed**| 25 min 37 s / 80.29         | 27 min 57 s / 67.73          | 11 min 27 s / 85.19      | 84 min 17 s / 82.15        |
>
> ------------------------
> ### **Weakness 3: SoTA model comparison.**
> Regarding broader comparisons, **there is currently limited prior work on synthetic data–driven, domain-specific KIE on document understanding area**. Existing RAG-style systems mainly address document-level retrieval or multi-page localization, rather than fine-grained textline or entity-level extraction. As such, direct baselines are scarce.
>
> ------------------------
> ### **Weakness 4: Ablation study on the impact of the two semantic adaptation tasks.**
> As the effectiveness of the joint-grained framework in VRDU has been demonstrated in prior work, our focus here is to evaluate the impact of synthetic datasets on semantic adaptation and structural adaptation. Following your suggestion, we include the results table below in the Appendix. Overall, the joint-grained design exhibits superior adaptation ability across most datasets. The only exception is EPHOIE, where performance gains are less pronounced—likely due to the limited Chinese text representation capability of the adopted pretrained layout backbone (LayoutLMv3-Chinese).
>
> | **Grainularity**                         | **FormNLU-P** | **FormNLU-H** | **CORD** | **EPHOIE** |
> |-------------------------------------------|--------------:|--------------:|---------:|-----------:|
> | **Fine-grained Span-based QA Head**       | 38.72 | 15.50 | 47.21 | 20.52 |
> | **Coarse-grained Entity Retrieving Head** | 65.35 | 25.55 | 63.17 | 29.74 |
> | **Joint (Fine + Coarse)**                 | 66.19 | 28.29 | 66.25 | 27.16 |
>
> ------------------------
> ### **Question 1: Table 1 Iterative Time Setting.**
> For Table 1, we report the best-performing iteration for each dataset: iteration 2 for FormNLU-P, FormNLU-H, and CORD, and iteration 4 for EPHOIE.

---

> > ### Comment · Reviewer_ksg5 · 2025-11-25
> > **Acknowledgment of rebuttal**
> >
> > Thanks to the authors for their detailed response. Their comments have helped to clarify my questions about the description of the method and the ablation study clarifies the role of the semantic adaptation.
> >
> > However the results on computation overhead do not resolve my concerns about the trade-off between accuracy and computation time of the proposed method. The proposed method only achieves slight improvement, adding complexity (extra-modules that have to be trained) and signifficant computation overhead. Therefore, It is not clear its advantage in comparison to using a plain MLMM.
> >
> > Regarding the comparison with SOTA for each dataset, in the literature there are several methods for KIE whose already published results could be included in the comparison.
> >
> > I will keep my original review since these two important points have not been resolved.

---

### Official Review · Reviewer_mKsU · 2025-10-31

**Soundness:** 2
**Presentation:** 2
**Contribution:** 2
**Rating:** 6
**Confidence:** 4

**Summary:**

The paper presents SynDoc, a unified framework for domain-specific Visually Rich Document Understanding (VRDU) that integrates discriminative and generative models to overcome hallucinations and poor domain adaptation in existing LLMs/MLLMs. SynDoc introduces a synthetic data generation pipeline that leverages document structure and domain-specific queries to create high-quality annotations, along with adaptive instruction tuning to enhance domain knowledge extraction. A recursive inference mechanism further refines model outputs for stable, accurate predictions. Experiments show that SynDoc achieves scalable and precise understanding across specialized domains such as medicine, finance, and material science.

**Strengths:**

- The paper introduces a scalable pipeline that automatically produces high-quality, domain-specific annotated data, reducing dependence on costly manual labeling.
- Adaptive Instruction Tuning is proposed, which enhances the discriminative model’s domain adaptation and knowledge extraction through targeted, domain-aware instruction tuning.
- The authors design a Recursive Inference Mechanism, which integrates discriminative and generative reasoning in a feedback loop, achieving more stable, accurate, and interpretable document understanding results across domains.

**Weaknesses:**

- The idea of integrating discriminative and generative models is good; however, the overall pipeline is somewhat engineered, which weakens the novelty of the proposed method.
- Besides the improved performance, the computation and storage overload is ignored in this manuscript, which is important for application is real-world system.

**Questions:**

Please refer to the weakness part.

---

> ### Author Response · Authors · 2025-11-22
>
> ### **Weakness 1: The overall pipeline is somewhat engineered, which weakens the novelty of the proposed method.**
> We understand that the modular structure of SynDoc may give the impression of an engineering-driven pipeline. While SynDoc does employ interoperable components, the contribution extends beyond engineering and introduces a conceptual framing for domain adaptation in multimodal document understanding.
>
> Specifically, SynDoc proposes a principled domain-scaling framework rather than a procedural assembly of models. The core novelty lies in (1) **shifting from traditional general-knowledge scaling in LLMs to domain-focused scaling**, and (2) **formalizing the role of a Joint-Grained Warmer as a domain-grounded retriever that replaces heavy generative model tuning**. The Warmer is not an auxiliary engineering step; it embodies the central hypothesis that discriminative domain grounding can precede and guide generative reasoning, reducing reliance on instruction tuning and manual labels.
>
> This research direction aligns with several strands of emerging ICLR-relevant work where architectural restructuring, retriever-first learning, and domain-specific scaling represent meaningful advances, rather than simple engineering refinements. The resulting paradigm demonstrates not only strong empirical performance but also a conceptual shift: domain adaptation does not necessarily require scaling the model, but scaling the domain knowledge efficiently.
>
> To help prevent misunderstanding, we will refine the exposition to emphasize:
> - the methodological insight behind domain-focused synthetic scaling
> - the theoretical motivation for training a domain-aware retriever before generative reasoning
> - and how the framework generalizes beyond the specific document setting tested in the paper.
>
> We hope these clarifications make the distinction between a purely engineered pipeline and the intended conceptual contribution more evident.
>
> -------------------------------
> ### **Weakness 2: Computation and Storage overload.**
> Our Warmer is built on a pretrained vision–language model of relatively small size (~140M parameters, ≈0.6 GB). As shown in Table 10 in the Appendix, training the Warmer requires approximately 25 GB of GPU memory when using a batch size of 16. Compared with fine-tuning large LLMs—which often demand substantially more GPU memory, longer training time, and higher computational cost—tuning the Warmer remains considerably more efficient in both computation and storage.

---

### Official Review · Reviewer_LTVw · 2025-11-03

**Soundness:** 4
**Presentation:** 4
**Contribution:** 4
**Rating:** 6
**Confidence:** 3

**Summary:**

The authors introduce SynDoc, a novel method for improving information extraction from documents. In this approach, they have a generative MLLM and discriminative warmer model. They use synthetic data generation pipeline to train the warmer, which then provides hints to the MLLM in a recursive inference loop.

**Strengths:**

- One of the main strength is the  synthetic data generation pipeline, which solves the labelled data bottleneck problem in visually rich documents.
- Their approach of using a discriminator to provide hints to the MLLM is very interesting and useful.

**Weaknesses:**

- The authors did not discuss about latency. Given that their approach involves recursive inferencing at test time, the latency will significantly increase.

**Questions:**

- The authors provide zero-shot comparison, in which their approach improved Gemini performance. It would be interesting to see how this would compare to fine-tune models. Can it improve performance of fine-tuned models as well?
- The approach depends on the data generator and warmer model. This can result in bias amplification, example where the warmer learns the generators flaws and reinforces those biases.

---

> ### Author Response · Authors · 2025-11-22
>
> ### **Weakness 1: The authors did not discuss about latency.**
> Thank you for raising this important point. We agree that recursive inference introduces additional latency. However, in practice, the overhead is modest for the following reasons:
>
> 1) **Minimal additional inference time.**
> Empirically, most queries converge within one iteration, and only a small fraction requires further refinement. As a result, the latency increase is limited in real scenarios. We will include detailed inference-time measurements in the appendix to make this clear.
>
> 2) **Substantial savings in training cost and data requirements.**
> SynDoc avoids costly LLM fine-tuning and the need for large annotated datasets, yet still achieves strong grounding performance. This shift—reducing heavy training requirements while adding only lightweight inference—offers a far more practical and resource-efficient alternative. This is especially important because fine-tuning MLLMs can introduce knowledge-leakage or knowledge-conflicting behaviours, which our approach avoids.
> | **Config**        | **FormNLU-P (Time / ANLS)** | **FormNLU-H (Time / ANLS)** | **CORD (Time / ANLS)** | **EPHOIE (Time / ANLS)** |
> |-------------------|-----------------------------|------------------------------|--------------------------|----------------------------|
> | **Vanilla**       | 15 min 24 s / 76.09         | 17 min 22 s / 66.86          | 6 min 26 s / 84.35       | 35 min 25 s / 81.82        |
> | **Best Performed**| 25 min 37 s / 80.29         | 27 min 57 s / 67.73          | 11 min 27 s / 85.19      | 84 min 17 s / 82.15        |
>
> We will add a discussion on the accuracy–latency trade-off in the revised manuscript and provide a supporting table in the appendix.
>
> --------------------------------
> ### **Question 1: Can it improve performance of fine-tuned models as well?**
> Thank you for sharing this compelling extension of our work. The use of synthetic data can help bridge the domain gap between pretrained backbones and the target domain. Introducing a synthetic data–based pretraining stage prior to fine-tuning may reduce dependence on high-quality manually annotated datasets, benefiting both pretrained vision–language models (e.g., LayoutLMv3, LiLT) and MLLMs. This adaptation possibly can facilitate faster convergence, improve performance, and further decrease the need for manual annotations. We definitely will consider this a promising direction for future research.
>
> -------
> ### **Question 2: Risk of bias amplification.**
> It is possible that the warmer introduces noise into the MLLM. As noted in our prompt in Appendix Table 9, we emphasied the given tips *"which may be incorrect"*. Ultimately, the MLLM relies on its implicit knowledge to determine how to utilize the retrieved content. Retrieval may primarily serve to direct the model's attention toward relevant information (Tips). For instance, in our Top-K setting, feeding the top-K retrieved segments—despite containing more noisy or incorrect content—can still lead to improved performance. With continued advances in document-structure parsing and synthetic-data generation workflows, this paradigm is expected to become increasingly robust.

---

### Author Response · Authors · 2025-12-03

Dear Area Chairs and Senior Area Chairs,

Thank you sincerely for your time, effort, and careful evaluation of our submission. We provide a concise overview of the conceptual contributions and a summary of the main strengths highlighted across reviews.

### **Overview of Contributions**
This work presents SynDoc, a new **paradigm** for domain-specific document Key Information Extraction (KIE) by shifting from
general-purpose LLM scaling to domain-specific adaptation tailored for concrete application scenarios. SynDoc introduces three core conceptual innovations:
- **Domain-scaled synthetic supervision**: A principled workflow that generates large quantities of structurally faithful, high-quality synthetic annotations to capture domain-specific document behaviours without manual labeling.

- **The “Warmer” mechanism**: A lightweight discriminative retriever that injects grounded multimodal and domain knowledge into MLLMs. This component enables iterative, synergistic refinement between retrieval and generation—without relying on expensive or risky MLLM fine-tuning.

- **Joint-grained domain adaptation**: A dual-level (fine-grained and coarse-grained) structural–semantic alignment strategy that bridges the gap between generic pretrained knowledge and domain-specific document patterns for more robust retrieving.

Together, these innovations form a scalable, annotation-efficient, and domain-adaptive framework, offering a new perspective on how multimodal LLMs can be effectively specialised for real-world document understanding tasks.

### **Main Strengths** (as consistently noted by reviewers)

- **Addresses real challenges in domain-specific VRDU:** Reviewers (LTVw, mKsU, ksg5, ntMK)  consistently acknowledged that SynDoc effectively tackles the **labelled-data bottleneck** in visually rich documents and has strong potential to generalise to other specialised domains (mKsU).

- **Novel and effective hybrid design (discriminative retriever + generative MLLM)**
The Warmer–MLLM collaboration was widely highlighted as an interesting (LTVw), original (ntMK) and practically useful idea (LTVw), improving grounding, interpretability, and domain adaptation (mKsU). Domain adaptive tuning strategies are introduced to enable the discriminative warmer capturing more domain specific knowledge (mKsU, ksg5).

- **Recursive inferencing for stable and interpretable predictions**
Reviewers noted that the recursive refinement loop meaningfully integrates discriminative and generative reasoning, leading to more stable, accurate, and interpretable document understanding across domains (mKsU). The creativity of the designing is mentioned by the reviewer (ntMK)

Across the reviews, the majority of reviewers acknowledged the creativity, effectiveness, and conceptual interest of the SynDoc framework.

---

### Author Response · Authors · 2025-12-03

### **Concerns and Response**

Reviewers raised several concerns about the work. Below, we provide concise responses along with the locations of the corresponding updates in the revised manuscript.
- **Latency Issue** All reviewers raised concerns about latency. We added Table S6 (Appendix D.2) to report detailed runtimes. While iterative refinement increases latency, the warmer injects domain-specific grounding into the MLLM, improving both performance and interpretability. Additionally, as shown in Table S8 (Appendix E.2), iterative prompting methods such as Self-Consistency and Reflexion require much longer inference yet still perform worse than SynDoc. This demonstrates that SynDoc achieves a stronger balance between latency and accuracy while maintaining better grounding efficiency.


- **Other SoTAs** As admitted by the reviewers (LTVw, ntMK) , SynDoc firstly introduces a new paradigm for enhancing MLLMs in domain-specific document understanding. Thus, there are no directly comparable models available. Nevertheless, we added Table S8 (as below) to include additional comparisons with widely used prompting methods and RAG-based frameworks including both performance and time latency. The results show that SynDoc effectively integrates domain-specific knowledge from the adapted warmer, achieving improved grounding and interpretability with only limited latency. Further analysis is provided in Appendix E.2.
| Model             | F-P ANLS | F-P Time   | F-H ANLS | F-H Time   | CORD ANLS | CORD Time | Ephoie ANLS | Ephoie Time |
|-------------------|----------|------------|----------|------------|-----------|-----------|--------------|--------------|
| CoT               | 77.78    | 15 min 11 s | 65.67    | 17 min 48 s | 84.56     | 6 min 23 s | 81.17        | 35 min 25 s  |
| Self-Consistency  | 75.78    | 44 min 59 s | 67.56    | 72 min 51 s | 83.99     | 33 min 11 s | 81.52        | 174 min 11 s|
| Reflexion         | 78.43    | 45 min 21 s | **68.19**    | 75 min 13 s | 85.11     | 40 min 32 s | 81.23        | 191 min 45 s |
| TF-IDF            | 75.24    | 15 min 02 s | 64.12    | 16 min 30 s | 83.22     | 6 min 01 s | 81.01        | 33 min 11 s  |
| BM25              | 75.45    | 16 min 34 s | 64.72    | 17 min 21 s | 83.48     | 6 min 32 s | 80.82        | 37 min 32 s  |
| Dense             | 74.25    | 15 min 43 s | 65.95    | 17 min 44 s | 83.51     | 6 min 22 s | 81.74        | 34 min 24 s |
| **SynDoc**        | **80.29** | 30 min 47 s | 67.73 | 31 min 17 s | **85.19** | 11 min 27 s | **82.15** | 84 min 17 s|


- **More Ablation Testing of Warmers**: We have expanded the ablation studies as requested. In addition to the results shown in Table 2, we added a new table in the Appendix to compare different granularity configurations, with detailed explanations provided in Appendix E.3 and Table S9. Since the effectiveness of the joint-grained framework has already been demonstrated, we did not originally include this breakdown in the main paper, but it is now provided for completeness.

- **The framework's novelty requires clearer articulation.** Reviewers ntMK and mKsU acknowledged the novelty and contributions of our framework but suggested improving the writing to avoid an overly engineering-focused presentation. Following their feedback, we revised the research aim and contribution statements (Introduction, L61–L94) to more clearly highlight the conceptual contributions of the framework.
- **Limited Performance Increasing** The reviewers ksg5 and ntMK raised concerns about the modest performance gains on some datasets. We acknowledge this limitation. As this work introduces a new paradigm, our goal is to demonstrate feasibility rather than fully optimise all components. The performance reflects an early-stage implementation, and further improvements in synthetic data, warmer training, and domain adaptation are expected. Importantly, this paradigm shifts VRDU toward scaling domain-specific knowledge, and—as noted by Reviewer mKsU—has clear potential for more interpretable and reliable gains as it evolves.
- **Typo and Writing updating**: Reviewers (LTVw and LTVw) agreed that the writing is generally clear with clear visualisations, though reviewer ksg5 noted some misalignment between the text and Figure 3 and requested clarification on certain framework design details. We have revised the method descriptions and updated the corresponding figure to ensure better alignment and clarity.

---

### Meta-Review · Area_Chair_9qbr · 2026-01-06

**Summary:**

Main concerns from the reviewers:
1. Empirical strength: Some reviewers consider the reported performance gains marginal compared to baseline MLLMs and question the accuracy-complexity trade offs.
2. Insufficient baselines: Some reviewers criticize the method is only compared with off-the-shelf MLLMs and request more SoTA baselines. The rebuttal argues there is no direct baselines to be compared with.
3. Clarity and ablation: While some reviewers think the paper is well presented, others pointed out substantial clarity/notation issues.
4. Latency/compute issues: Some reviewers ask for a clearer accuracy-latency trade-off. The rebuttal provides more results and explanations, which at least partially address the issue.

To sum up, while many of the concerns raised have been addressed in the rebuttal, I see two remain (or partially addressed): insufficient comparisons with more SoTA baselines and unclear cost-benefit trade-off. I think this is a borderline case, and acceptance would depend on the capacity of the acceptance rate.

**Reviewer Concerns:**

Concerns addressed:
1. Presentation related issues: This is mostly addressed by carefully revising the paper.
2. Additional ablations on Warmer: The rebuttal provides additional results in the appendix, which I believe is sufficient to address the issue.
3. Computation and storage: The rebuttal provides clarification on this by providing the exact setting.

Concerns un-resolved/partially resolved:
1. Latency-accuracy trade-off: Although the rebuttal provides additional results on this, one reviewer still considers this as slight improvement with significant computation overhead.
2. Comparisons with task-specific SoTA: Some reviewer ask for more comparisons, but the rebuttal consider there is no directly SoTA model to be compared with. The reviewer did not seem to be satisfied with this. Although the reviewer did not point out which baseline, I think one candidate could be the LayoutLM or just the proposed method without the Warmer->MLLM loop. Adding these comparisons would strengthen the paper.

**Reviewer Scores:**

Based on the reviews and rebuttal, I consider the positive reviewers would remain their score (6), while the negative ones could increase the scores by 1-2 points but would still remain negative.

---

### Decision · Program_Chairs · 2026-01-26

Reject